# State fusion entropy for continuous and site-specific analysis of landslide stability changing regularities

Yong Liu[1], Zhimeng Qin[1], Baodan Hu[1], Shuai Feng[1]

[1]School of Mechanical Engineering and Electronic Information, China University of Geosciences, Wuhan, 430074, China

*Correspondence to*: Zhimeng Qin (imqinzhimeng@163.com)

**Abstract.** Stability analysis is of great significance to landslide hazard prevention, especially the dynamic stability. However, many existing stability analysis methods are difficult to analyse the continuous landslide stability and its changing regularities in a uniform criterion due to the unique landslide geological conditions. Based on the relationship between displacement monitoring data, deformation states and landslide stability, a state fusion entropy method is herein proposed to derive landslide

instability through a comprehensive multi-attribute entropy analysis of deformation states which are defined by a proposed joint clustering method combining K-means and cloud model. Taking Xintan landslide as the detailed case study, cumulative state fusion entropy presents an obvious increasing trend after the landslide entered accelerative deformation stage and historical maxima match highly with landslide macroscopic deformation behaviours in key time nodes. Reasonable results are also obtained in its application to several other landslides in the Three Gorges Reservoir in China. Combined with field survey,

state fusion entropy may serve for assessing landslide stability and judging landslide evolutionary stages.

## 1 Introduction

Landslide is one of the major natural hazards, accounting for massive damages of properties every year (Dai et al., 2002). Analysis of landslide stability as well as its changing regularities plays a significant role in risk assessment at site-specific landslides (Wang et al., 2014). For this concern, many stability analysis methods have been proposed, such as Saito's method,

limit equilibrium method (LEM) and finite element method (FEM) (Saito, 1965; Duncan, 1996; Griffiths and Fenton, 2004). Saito's method is an empirical forecast model and is suitable for the prediction of sliding tendency and then the failure time. Based on homogeneous soil creep theory and displacement curve, it divides displacement creep curves into three stages: deceleration creep, stable creep and accelerating creep, and establishes a differential equation for accelerating creep. The physical basis of Saito's method helped it to successfully forecast a landslide that occurred in Japan in December 1960, but

also makes it strongly dependent on field observations. LEM is a kind of calculation method to evaluate landslide stability based on mechanical balance principle. By assuming a potential sliding surface and slicing the sliding body on the potential sliding surface firstly, LEM calculates the shear resistance and the shear force of each slice along the potential sliding surface and defines their ratio as the safety factor to describe landslide stability. LEM is simple and can directly analyse landslide stability under limit condition without geotechnical constitutive analysis. However, this neglect of geotechnical constitutive

characteristic also restricts it to a static mechanics evaluation model that is incapable to evaluate the changing regularities of landslide stability. In the meanwhile, LEM involves too many physical parameters such as cohesive strength and friction angle, which makes it greatly limited in landslide forecast. As a typical numerical simulation method, FEM subdivides a large problem into smaller, simpler parts that are called finite elements. The simple equations that model these finite elements are then assembled into a larger system of equations that models the entire problem. FEM then uses variational methods from the calculus of variations to approximate a solution by minimizing an associated error function. In landslide stability analysis, FEM can not only satisfy the static equilibrium condition and the geotechnical constitutive characteristic, but also adapt to the discontinuity and heterogeneity of the rock mass. However, FEM is quite sensitive to various involved parameters and the computation will increase greatly to get more accurate results. If parameters and boundaries are precisely determined, LEM and FEM can provide results with high reliability. Other stability analysis methods such as strength reduction method also have been rapidly applied (Dawson et al., 2015). These methods provide the theoretical basis for analysing landslide stability and have been widely applied in engineering geology (Knappett, 2008; Morales-Esteban et al., 2015).

Despite of the great contributions made by these stability analysis methods, there are a few matters cannot be neglected. Firstly, safety factor is the most adopted index to indicate landslide stability (Hsu and Chien, 2016), but it mainly indicates safe (larger than 1) or unsafe (smaller than 1), incapable to show the degree of stability or instability (Li et al., 2009; Singh et al., 2012). Secondly, external factors such as rainfall (Priest et al., 2011; Bernardie et al., 2015; Liu et al., 2016) and fluctuation of water level (Ashland et al., 2006; Huang et al., 2017b) will also change landslide stability. But for now only a few literatures mentioned real-time landslide stability (Montrasio et al., 2011; Chen et al., 2014). Thirdly, methods like LEM and FEM involve too many physical parameters whose uncertainties make these methods hard to match with the real-time conditions of landslide. It becomes of great interest to find a new method to evaluate landslide stability, which only requires a few parameters, easily be matched with landslide real-time conditions, and can indicate the extent as well as the changing regularities of landslide stability.

Displacement is the most direct and continuous manifestation of landslide deformation promoted by external factors and has been widely used in landslide analysis (Asch et al., 2009; Manconi and Giordan, 2015; Huang et al., 2017a). Due to its easy acquisition, quantification and high reliability, displacement monitoring data has become one of the most recognized evidence for landslide stability analysis and early warning. Macciotta et al. (2016) suggested that velocity threshold be used as a criterion for early warning system and the annual horizontal displacement threshold for Ripley Landslide (GPS 1) can be 90 mm and that between May and September can be 25 mm. Based on the analysis of a large number of displacement monitoring data, Xu and Zeng (2009) proposed that deformation acceleration be used as an indicator of landslide warning, and the acceleration threshold of Jimingsi landslide was regarded as 0.45 mm/d$^2$ and that of another landslide in Daye Iron Mine as 0.2 mm/d$^2$. Federico et al. (2012) presented a systematic introduction to the prediction of landslide failure time according to the displacement data. However, although displacement data has been widely used in landslide analysis, it is hard to define a unified displacement threshold due to the unique geological conditions and many studies draw their conclusions directly based on original data and personal engineering geological experience.

Entropy has been widely used to describe the disorder, imbalance, and uncertainty of a system (Montesarchio et al., 2011; Ridolfi et al., 2011). Previous works have introduced entropy into landslide susceptibility mapping to evaluate the weights of indexes (Pourghasemi et al., 2012; Devkota et al., 2013). In the viewpoint of system theory, a landslide can be regarded as an open system and exchanges energy and information with external factors. Shi and Jin (2009) proposed a generalized

information entropy approach (GIE) to evaluate the "energy" of multi-triggers of landslide and found that the GIE index showed a mutation before landslide failure in a case study. But this GIE method is aimed at landslide triggering factors and thus cannot directly indicate landslide stability.

In this paper, a state fusion entropy approach is proposed for continuous and site-specific analysis of landslide stability changing regularities. It firstly defines deformation states as an integrated numerical feature of landslide deformation.

Considering the multiple attributes of deformation states, entropy is adopted for landslide stability (instability) analysis. Correspondingly, a historical maximum index is introduced to identify key time nodes of stability changes.

## 2 Methods

In this paper, landslide is regarded as an open dynamic system, and landslide stability (instability) is the source of the system. Under the influence of external factors, landslide stability will respond to these triggers by generating deformation states.

Eventually, deformation states will be manifested in the form of landslide displacement. Therefore, to analyse landslide stability based on displacement monitoring data, defining deformation states is the primary foundation. In order to adapt to the unique geological conditions of different landslides, a joint clustering method combining K-means clustering and cloud model is proposed. Aiming at three typical characteristics of deformation states, entropy analysis is then conducted and fused to analyse landslide instability and its changing regularities. Result interpretation method is proposed correspondingly. The flow

chart is shown in Figure 1.

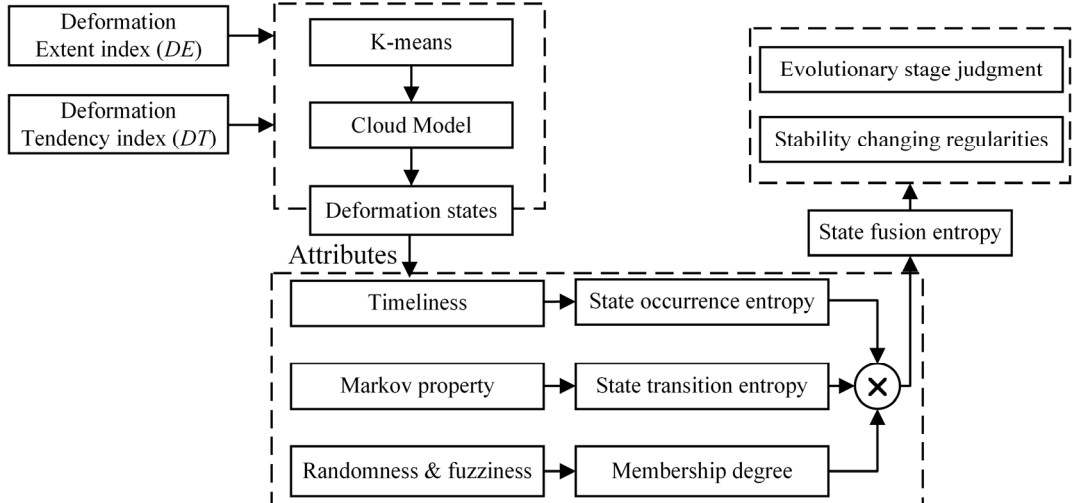

**Figure 1**. Flow chart of state fusion entropy method

## 2.1 Deformation state definition based on K-means combined with Cloud Model

Many deformation states exist during the development of landslide (Wu et al., 2016) and link up landslide stability and displacement monitoring data. On the one hand, deformation states indicate temporary landslide stability. On the other hand, deformation states can be manifested by displacement monitoring data. Therefore, the excavation of deformation states can be the primary step for analysing landslide stability analysis and its changing regularities according to displacement data. Due to the unique geological conditions of different landslides, a unified definition of deformation states seems infeasible. In view of this, the data-driven K-means clustering method and cloud model are integrated to investigate deformation states.

K-means is one kind of unsupervised clustering methods of vector quantization and is popular in data mining. It aims to partition $N$ observations into $K$ clusters in which each observation belongs to the cluster with the nearest mean (Steinley, 2006; Hartigan and Wong, 2013). Given a set of observations $(x_1, x_2, \ldots, x_N)$, where each observation is a $d$-dimensional real vector, K-means clustering aims to partition the $N$ observations into $K(K \leq N)$ sets $S = \{S_1, S_2, \ldots, S_K\}$. Formally, the objective is to find the $K$ sets to minimize intra-class distance and maximize inter-class distance through iterations. The objective of K-means can be expressed as eq. (1).

$$arg \min_S D_{intra} = arg \min_S \sum_{i=1}^{K} \sum_{x \in S_i} |x - c_i|^2$$
$$arg \min_S D_{inter} = arg \max_S \sum_{i=1}^{K} \sum_{j=1}^{K} |c_i - c_j|^2 \tag{1}$$

where $c_i$ is the mean of points in $S_i$; $D_{intra}$ is the pairwise squared deviations of points in the same cluster, representing the consistency of each cluster; $D_{inter}$ is the squared deviations between points in different clusters, reflecting the differences among clusters.

K-means clustering method is simple, fast and efficient. All observations will be labeled after clustering. However, since the clustering process is unsupervised, the cluster labels of observations are unstable and have a certain randomness. In the meanwhile, K-means algorithm lacks the index to distinguish observations in the same cluster, which leads to high fuzziness of cluster labels. Aiming at the randomness and fuzziness of cluster labels, cloud model is introduced to offer help.

Cloud model was proposed in 1995 to analyze the uncertain transformation between qualitative concepts and their quantitative expressions (Li et al., 1995). Among all cloud models, the normal cloud model is most popular one due to its universality (Li and Liu, 2004). Let $U$ be a universe of quantitative values, and $C$ be the qualitative concept of $U$. For any element $x$ in $U$, if there exists a random number $y = \mu_A(x), y \in [0, 1]$ with a stable tendency, then $y$ is defined as the membership (certainty) of $x$ to $C$ and the distribution of $y$ on the universe $U$ is defined as a cloud. Cloud model uses the expectation ($E$), entropy ($En$) and hyper-entropy ($He$) to characterize a qualitative concept, and integrates the ambiguity and randomness of the concept. Expectation is the central value of the concept in the universe, and is the value that best represents the qualitative concept. Entropy reflects the ambiguity of the qualitative concept and indicates the range of values that the concept accepts in the universe. Hyper-entropy indicates the randomness of membership. The diagram of digital features of one-dimensional cloud is shown in Figure 2. Given the digital features of a one-dimensional normal cloud [*Ex*, *Enx*, *Hex*], cloud droplets can be generated by forward cloud generator (*CG*) in the following orders. 1) Generate a normal random number $x$ with *Ex* as the

mean and *Enx* as variance; 2) Generate a normal random number *Enx'* with *Enx* as the mean and *Hex* as variance; 3) Calculate the membership as eq. (2) and each (x, y) is defined as a cloud droplet; 4) Repeat the above steps until required number of cloud droplets are generated. Correspondingly, the process of calculating digital features based on cloud droplets is called the backward cloud generator ($CG^{-1}$).

$$5 \quad y = \exp\left\{-\frac{(x-Ex)^2}{2Enx'^2}\right\} \tag{2}$$

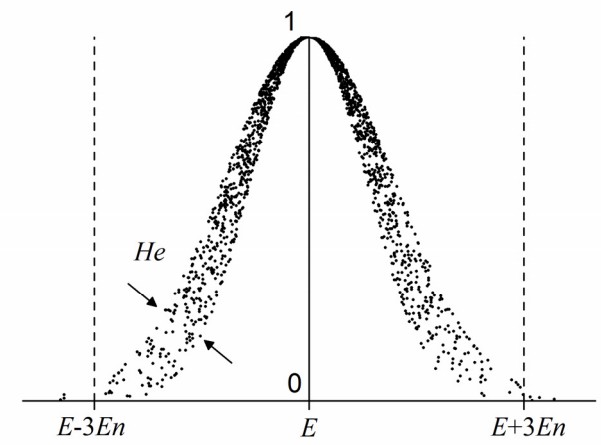

**Figure 2.** Digital features of one-dimensional cloud

K-means can automatically derive labels (concepts) from data but cannot distinguish items with the same label. Cloud model can utilize the distribution characteristics of data and express the membership of each data item to corresponding concept, but

10 cannot work without defining concepts. Therefore, a joint clustering method combining K-means and cloud model is proposed to define landslide deformation states according to displacement monitoring data. To describe clearly this method, two functional data type are defined for landslide displacement data. One is to indicate deformation extent (*DE*) and the other to deformation tendency (*DT*). Positive *DT* indicates an increasing deformation. The process of defining deformation states is as follows.

15       Step 0. Unite *DE* and *DT* at the same time as an item, i.e., (*DE*, *DT*);

      Step 1. Cluster all items based on K-means and obtain cluster labels (K_label) and the distance of each item to corresponding cluster centroid ($d_{ic}$);

      Step 2. For each cluster (cloud)

          a) Select a proportion of items as the typical items based on $d_{ic}$;

20           b) Conduct backward cloud generator ($CG^{-1}$) on typical items to obtain the digital features of this cloud;

          c) Conduct forward cloud generator (*CG*) to generate cloud droplets based on the digital features for visual analysis;

      Step 3. Calculate and normalize the memberships of each item to all clouds, and define the cloud label with the largest membership as the deformation state of corresponding item.

As can be seen from above procedures, the definition of deformation states is basically driven by displacement monitoring data and thus can adapt the unique geological conditions of different landslides. In the meanwhile, membership can be used to distinguish the displacement data with the same deformation state. Since displacement data is acquired in chronological order, the result is also a time-related state sequence.

As for the deformation state sequence, three typical attributes need to be noticed, respectively the timeliness, the Markov property and fuzziness. Timeliness is the primary attribute of each deformation state and is the basis of stability analysis. The Markov property is caused by the continuity and hysteresis characteristic of external triging factors such as rainfall and fluctuation of water level (Bordoni et al., 2015). The fuzziness is introduced in the process of defining deformation states.

## 2.2 Fusion entropy analysis of deformation state sequence

Entropy is an indicator of the degree of system chaos. Introduced in communication system by Shannon in 1948, entropy has become the basis of information theory (Shannon, 1948). Let $X$ be a discrete random variable, $x$ is one state of $X$, $p(x)$ is the probability when $X = x$. The information entropy of $X$ can be calculated by eq. (3).

$$I(x) = -\log p(x)$$
$$H(X) = \sum_{x \in X} p(x) \cdot I(x)$$
(3)

where $I(x)$ is the information amount of $x$; $H(X)$ is the entropy of $X$. As shown in eq. (3), information amount increases with

the decrease of probability. $H(X)$ is the statistical average of the information amount of each state, representing the overall uncertainty of $X$. The $p(x)$-weighted·$I(x)$ can be regarded as the individuation of state $x$ to overall uncertainty $H(X)$.

As for landslide deformation states, each of them contains some information about landslide stability. Slight deformation occurs frequently but indicates a relatively stable state of landslide. Severe deformation occurs rarely but indicates a really high instability of landslide and should draw the high attention for early warning. Therefore, entropy analysis is conducted to

analyse landslide instability based on deformation states. Aiming at the timeliness and Markov property of deformation state sequence, state occurrence entropy and state transition entropy are defined. Eventually, the product of state occurrence entropy, state transition entropy and membership is defined as the state fusion entropy to describe the comprehensive information about landslide instability.

State occurrence entropy (SOE) mainly aims to measure the information about landslide stability provided by a single

occurrence of one deformation state. Considering the great significance of severe deformation to landslide early warning, the basic equation of information entropy is modified to emphasize the probability difference between severe and slight deformation. In the meanwhile, to show the deformation tendency, the sign of state occurrence entropy is defined to be the same as $DT$, which also reflects the timeliness of deformation states. State occurrence entropy is defined as eq. (4).

$$SOE_{i,t} = \frac{-\log(p_i)/N_i}{\sum_{i=1}^{K} -\log(p_i)/N_i} \cdot sign(DT_t)$$
(4)

where $p_i$ is the probability of deformation state $i$; $N_i$ is the frequency of deformation state $i$; $K$ is the number of deformation states, i.e., the cluster number in K-means clustering method; $DT_t$ is the deformation tendency index ($DT$) at time $t$; $SOE_{i,t}$ is the state occurrence entropy of the occurrence of state $i$ at time $t$.

State transition entropy ($STE$) focuses on the measurement of the information about landslide stability when one deformation state transmits to another. Markov property describes such a property of a discrete state sequence that each state is only influenced by the former one state, independent to other states (Tauchen, 1986). Because the influence of external factors on landslide has the continuity and hysteresis characteristic, deformation state sequence satisfies the Markov property. Therefore, the state transition matrix of Markov Chain is employed to quantitatively analyze the transition regularities of deformation states. State transition entropy is defined as eq. (5).

$$STE_{ij} = \frac{-p_{ij} \cdot \log(p_{ij})}{\sum_{j=1}^{K} -p_{ij} \cdot \log(p_{ij})} \tag{5}$$

where $p_{ij}$ is the transition probability from former state $i$ to current state $j$; $K$ is the number of deformation states, i.e., the cluster number in K-means clustering method; $STE_{ij}$ is the state transition entropy of the transition from former state $i$ to current state $j$. As for landslide deformation states, on the one hand, severe deformation occurs rarely, resulting in a small probability of transitions from other deformation states to severe deformation. On the other hand, severe deformation indicates a high instability of a landslide and thus has a characteristic of poor sustainability. Apparently, the longer the severe deformation lasts, the higher instability it indicates the landslide and the larger $STE$ will be.

Finally, state fusion entropy ($SFE$) is defined as the product of state occurrence entropy, state transition entropy and membership degree, as shown in eq. (6). This definition is mainly based on the following reasons: 1) although state occurrence entropy and state transition entropy emphasize the different attributes of deformation states, they are both expressed in the form of information entropy; 2) they share the common engineering significance that the larger the entropy, the higher instability the landslide; 3) the membership in cloud model indicates the extent that displacement data support the deformation state concept and thus deserves consideration. Essentially, state fusion entropy is the individual contribution of temporary deformation state to landslide overall instability. By accumulating state fusion entropy according to time, cumulative state fusion entropy (CSFE) can be obtained.

$$\begin{aligned} SFE_{j,t} &= SOE_{j,t} \cdot STE_{ij} \cdot MBS_j \\ CSFE_t &= \sum_{t_0}^{t} SFE_{j,t} \end{aligned} \tag{6}$$

## 2.3 Result interpretation of state fusion entropy

State fusion entropy ($SFE$) is the comprehensive representation of the timeliness, the Markov property and fuzziness attributes of deformation states. In mathematical form, state fusion entropy can be regarded as the weighted information amount, indicating the individuation of each deformation state to overall landslide instability. For the value, on the one hand, the sigh of state fusion entropy is determined by $DT$, indicating the deformation tendency of landslide. Positive $DT$ indicates a growing

instability, and negative *DT* indicates a decreasing instability. On the other hand, the instable extent is represented by the absolute value of state fusion entropy.

Cumulative state fusion entropy (CSFE) is the sum of state fusion entropy, as shown in eq. (6). According to information theory, entropy indicates the overall uncertainty and instability of source. Likewise, cumulative state fusion entropy reflects the overall instability of landslide in the whole monitoring period. In other words, cumulative state fusion entropy represents the cumulative effect of landslide instability. As time goes on, cumulative state fusion entropy will also indicate the changing regularities of landslide instability. If landslide stays in a slight deformation period, cumulative state fusion entropy will maintain at a relatively low level. If landslide develops into a severe deformation period, cumulative state fusion entropy will accordingly show a continuous growth. Besides, a historical maximum index is introduced to identify key time nodes of stability changes. It is defined as the maximum from the very beginning to the time in question of cumulative state fusion entropy. Each renewal of historical maximum suggests a more dangerous state of landslide. Once new historical maximum occurs frequently, the cumulative state fusion entropy curve will inevitably increase significantly, indicating a high instability of a landslide. In this case, field survey will be necessary for landslide early warning and hazard prevention.

## 3 Case study

To verify the effectiveness of the state fusion entropy method, five landslides in the Three Gorges Reservoir area in China were selected as examples for stability changing regularities analysis. Among them, Xintan landslide is a reactive landslide triggered by rainfall and has failed. Baishuihe landslide, Bazimen landslide and Shuping landslide are reactive landslides mainly triggered by reservoir water level and rainfall. Pajiayan landslide is a new-born landslide. Limited by space, results of Xintan landslide are detailed illustrated and that of others are simply presented.

Xintan landslide, which occurred 26.6 km upstream of the Three Gorges dam and 15.5 km downstream of Zigui County, is located in Xintan town on the north shore of Yangtze River. It extends from south to north with a length of 2000 m. The width of the rear edge is about 300 m and the width of the front edge is between 500 m and 1000 m, with an average width of 450 m. The elevation decreases from about 900 m in the north to 65 m in the south with an average gradient of about 23°. The main body of the deep-seated landslide is comprised of colluvial deposits overlying the bedrock of shale stone of Silurian system, sandstone of Devonian system and limestone of Carboniferous and Permian system. The strike of the bedrock strata is mainly N10°-30°E, almost perpendicular to the Yangtze River. At the end of 1977, a monitoring system of surface displacement composed of four collimation lines was set up and eight markers were added in July 1984 mainly by the Avalanche Survey Department of Xiling Gorge. Thanks to this monitoring and field investigation, the losses were controlled to the possible minima, without any fatalities and injuries when Xintan landslide failed on June 12, 1985 (Zhang et al., 2006; Huang et al., 2009; Lin et al., 2013). According to previous studies, cumulative horizontal displacements at A3 and B3 are considered to be the most representative (Wang, 2009). Location and two monitoring points of Xintan landslide are shown in

Figure 3. Monthly horizontal displacement of A3 from January 1978 to May 1985 is shown in Table 1. Since the displacement of A3 was obtained monthly, deformation states and state fusion entropy will also be monthly indexes for Xintan landslide.

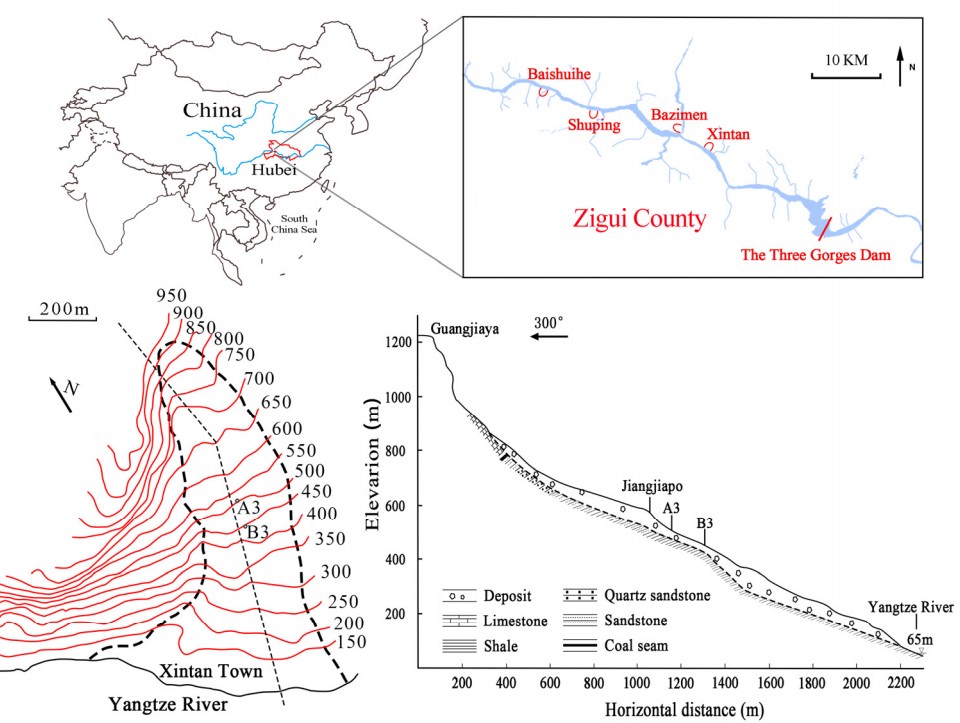

**Figure 3.** Location and plane/section of Xintan Landslide

**Table 1.** Monthly horizontal displacement of A3 from January 1978 to May 1985 (mm)

|      | Jan.  | Feb.  | Mar.  | Apr.  | May   | Jun.  | Jul.  | Aug.  | Sep.  | Oct.  | Nov.  | Dec.  |
|------|-------|-------|-------|-------|-------|-------|-------|-------|-------|-------|-------|-------|
| 1978 | 0.0   | 3.2   | 7.1   | 6.1   | 6.4   | 8.7   | 13.1  | 6.9   | 4.8   | 1.8   | 1.8   | 7.5   |
| 1979 | 1.6   | 7.2   | 1.0   | 1.6   | 5.0   | 10.5  | 5.7   | 19.7  | 336.3 | 161.3 | 39.8  | 1.6   |
| 1980 | 14.3  | 11.1  | 7.8   | 4.2   | 9.7   | 9.7   | 80.8  | 49.5  | 59.5  | 20.7  | 9.4   | 18.7  |
| 1981 | 22.9  | 6.9   | 6.2   | 10.6  | 6.5   | 5.0   | 3.4   | 10.2  | 8.6   | 11.7  | 6.8   | 1.9   |
| 1982 | 8.2   | 7.5   | 6.8   | 33.2  | 66.7  | 82.2  | 54.5  | 344.2 | 430.6 | 525.6 | 433.5 | 35.3  |
| 1983 | 45.3  | 15.0  | 31.8  | 16.4  | 20.0  | 20.8  | 43.9  | 348.1 | 101.3 | 171.2 | 298.7 | 156.2 |
| 1984 | 69.1  | 51.3  | 27.6  | 15.5  | 49.0  | 127.9 | 196.0 | 320.1 | 136.1 | 413.8 | 325.8 | 214.6 |
| 1985 | 142.1 | 146.1 | 153.3 | 123.0 | 296.1 |       |       |       |       |       |       |       |

Considering that the monitoring error of GPS can be ignored compared to landslide actual deformation on monthly time scale, monthly deformation velocity ($v$) was selected as the $DE$ index and monthly deformation acceleration ($a$) as the $DT$ index. Firstly, monthly deformation states were defined based on joint clustering method of K-means and Cloud Model with monthly deformation velocity and acceleration as the inputs. Given that there are about 90 monthly items with 2 dimensions, i.e., ($v$, $a$),

cluster number $K$ was empirically set to 3 for simplicity. The initial cluster centroids were determined by performing preliminary clustering phase on a random 10% subsample of data set. The clustering process was repeated 9 times and the cluster labels (K_label) were determined based on voting strategy. Cluster centroids and number of items in each cluster are shown in Table 2.

**Table 2.** Cluster centroids and number of items in each cluster

| K_label | Velocity ($v$) | Acceleration ($a$) | Items |
|---------|----------------|--------------------|-------|
| 1 | 30.83 | 0.13 | 69 |
| 2 | 133.53 | 201.86 | 8 |
| 3 | 366.75 | -133.09 | 10 |

As can be seen from Table 2, obvious numerical differences exist among cluster centroids, suggesting different deformation patterns. Most items belong to the first cluster, whose deformation velocity and acceleration maintain at a relative low level, proving the fact that the occurrence probability of slight deformation is large while that of severe deformation is small during the development of landslide.

Then cloud model continued to evaluate the membership of each item to corresponding cluster label. In view of the Non-negative numerical limit of deformation velocity, cluster 1 was set as a right half cloud, cluster 2 as a symmetric cloud and cluster 3 as a left half cloud in deformation velocity dimension. In deformation acceleration dimension, all clusters were set as symmetric clouds. The regenerated clouds is shown in Figure 4. After obtaining digital features of each cloud, membership of each item to all clouds were calculated and unified, and the cloud label (CM_label) with the largest membership was defined

as the monthly deformation state. Comparison of K_label and CM_label is shown in Figure 5. As can be seen, K_label and CM_label are almost the same. But there are some items which belonged to cluster 1 in K-means, now belong to cluster 2 or 3 in cloud model, indicating that cluster 1 has a small tolerance to numerical deviations.

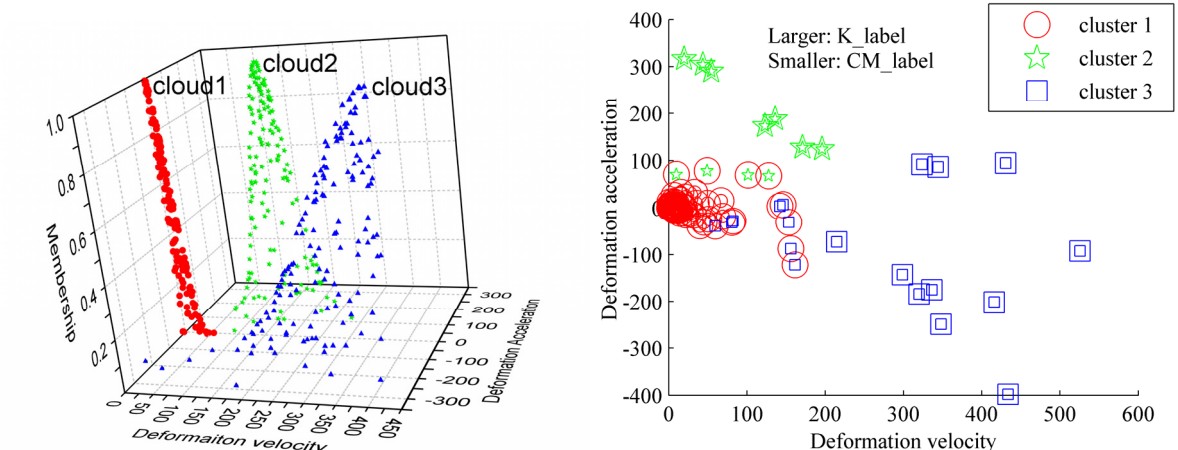

**Figure 4.** Regenerated clouds of each cluster       **Figure 5.** Comparison of K_label and CM_label

After the joint clustering process of K-means and Cloud Model, monthly deformation states were derived and after which state fusion entropy analysis of deformation state sequence was followed. As mentioned in the methods section, state occurrence

entropy and state transition entropy are defined aiming at the timeliness and Markov property of deformation states. After the statistics of the frequency and probability of each deformation state, state occurrence entropy of each deformation state was calculated based on eq. (4), whose absolute values were respectively 0.1621, 0.4980 and 0.3399. State transition matrix was obtained by analyzing deformation state sequence and state transition entropy obtained based on eq. (5) is shown in Table 3.

5    Three values are mainly discussed here: 1) the state transition entropy from S2 to S1 is zero. As mentioned earlier, S2 has a relatively large deformation velocity while S1 has a smaller one. So a deceleration process which corresponds to S3 will inevitably show up between S2 and S1; 2) the transition from slight deformation S1 to S1 presents a small transition entropy, indicating a small risk of landslide; 3) the maximum transition entropy occurs in the transition from S2 to S2, indicating an increasing instability.

10   **Table 3.** State transition entropy of Xintan Landslide

| Deformation state | S1 | S2 | S3 |
|:---:|:---:|:---:|:---:|
| S1 | 0.2679 | 0.4687 | 0.2634 |
| S2 | 0.0000 | 0.5516 | 0.4484 |
| S3 | 0.3635 | 0.3112 | 0.3253 |

Finally, monthly state fusion entropy was calculated based on eq. (6), as shown in Figure 6. Between December 1977 and December 1981, monthly state fusion entropy remains at a low level, fluctuating around zero. There are two local maxima but only last a short time. Between January 1982 and May 1982, values which are close to the local maxima in earlier stage occur frequently, indicating the increasing instability and higher risk of Xintan landslide.

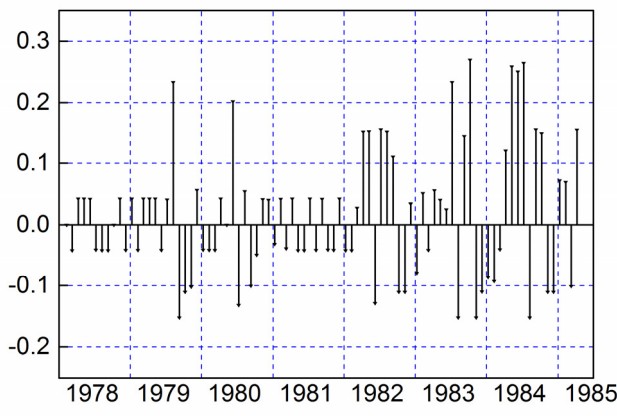

**Figure 6.** Monthly state fusion entropy of Xintan landslide

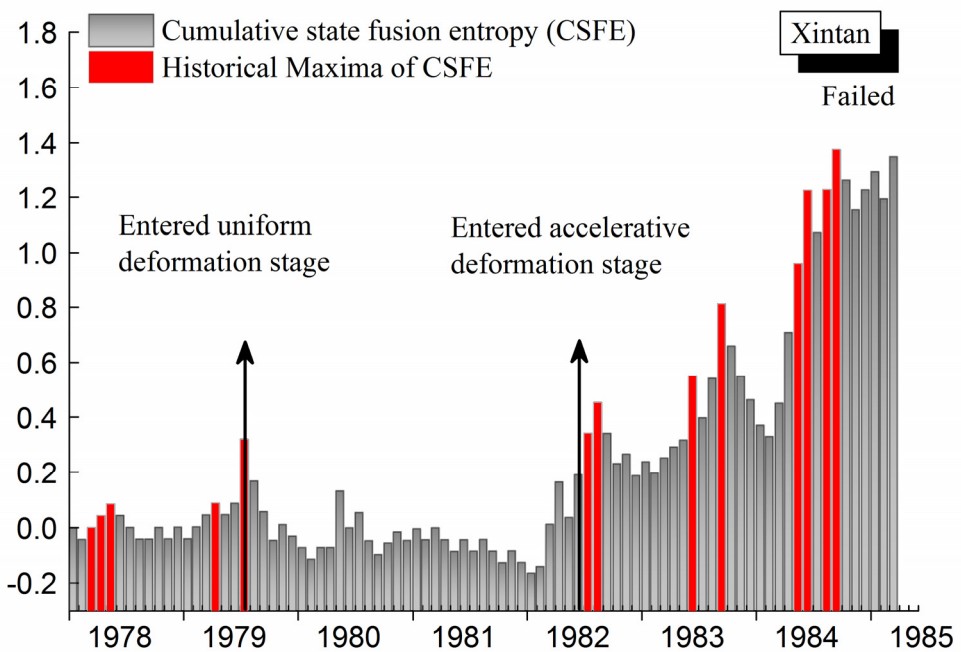

**Figure 7.** Cumulative state fusion entropy and historical maxima of Xintan landslide

For an insight into the cumulative effect and changing regularities of landslide instability, cumulative state fusion entropy was calculated, after which historical maxima were picked out, as shown in Figure 7. As for the cumulative state fusion entropy curve, there are two typical changing forms: fluctuation around zero type and fluctuant increasing type. The first type occurs between January 1978 and February 1982, during which the cumulative state fusion entropy fluctuates around zero with a slight decrease. A local maximum occurs in August 1979. The global minimum occurs in February 1982. After February 1982, cumulative state fusion entropy shows an apparent fluctuant increasing trend. Historical maxima mainly concentrate in two periods. From January 1978 to July 1979, the first period is at the prophase of monitoring period and the historical maximum is relatively small, easy to be updated. From June 1982 to April 1985, the second period is at the anaphase of monitoring period. During this time, the frequent renewal of historical maximum indicates actually an increasing instability of Xintan landslide and higher risk of landslide hazard.

The macroscopic behaviors of Xintan landslide near historical maxima were investigated according to previous studies (Wang, 1996). Around August 1982, the front edge of Jiangjiapo went through a small collapse. In June 1983, the colluvial deposits between Guangjiaya and Jiangjiapo showed signs of resurrection. At the end of 1984, the trailing edge of the landslide showed an "armchair" shape and the leading edge was bulged out. Some collapse pits were found on the upper side while several new tensile cracks in the middle. Meanwhile, some small collapses which seem irrelevant to rainfall occurred. In May 1985, old cracks widened and new cracks appeared, forming a ladder-shaped landing ridge. Moreover, Jiangjiapo presented a clear trend of the overall slippage. These proofs suggest that the historical maximum index is highly consistent with landslide macroscopic deformation behaviors.

Many studies have claimed the close relationship between landslide stability and evolutionary stages (Xu et al., 2008). And thus the evolutionary stages of Xintan landslide was introduced to verify the effectiveness of the state fusion entropy method. According to previous studies, Xintan landslide entered uniform deformation stage in August 1979, entered accelerative deformation stage in July 1982, and failed in June 1985 (Yin et al., 2002). As shown in Figure 7, August 1979 corresponds to a local mutation of cumulative state fusion entropy and is also the end of the first period of historical maxima. July 1982 is located at the fluctuant increasing period of cumulative state fusion entropy and it is the start of the second period of historical maxima. Before the failure of Xintan landslide, cumulative state fusion entropy has already reached a really high level in April 1985. In other words, historical maxima match really well with the evolutionary stages of Xintan landslide in key time nodes, and can suggest the effectiveness of this method. Furthermore, when Xintan landslide entered accelerative deformation stage in July 1982, cumulative state fusion entropy starts an obvious fluctuant increasing trend. In this aspect, the fluctuant increasing type of cumulative state fusion entropy may serve as a new clue to determine whether a landslide enter the accelerative deformation stage or not.

In addition, the results of several other studies were introduced for comparison. Chen (2014) studied the stability of Xintan landslide by FEM with consideration of the loading effect and material weakening caused by rainfall, as shown in Figure 8. Moreover, the result of an unloading-loading response ratio method (ULRR) is also introduced (Zhang et al., 2006; He et al., 2010), as shown in Figure 8. Because only annual results are given in these studies, annual average of CSFE were correspondingly calculated for comparison. According to Figure 8, the safety factor decreases year by year and cannot reflect the recovery process of landslides stability. The ULRR presents similar changing regularities like CSFE after Xintan landslide entered in accelerative deformation stage in 1982. But the mutation in 1981 when Xintan is still in uniform deformation stage seems unreasonable. Besides, ULRR is obtained yearly and offers less details about stability changes than CSFE.

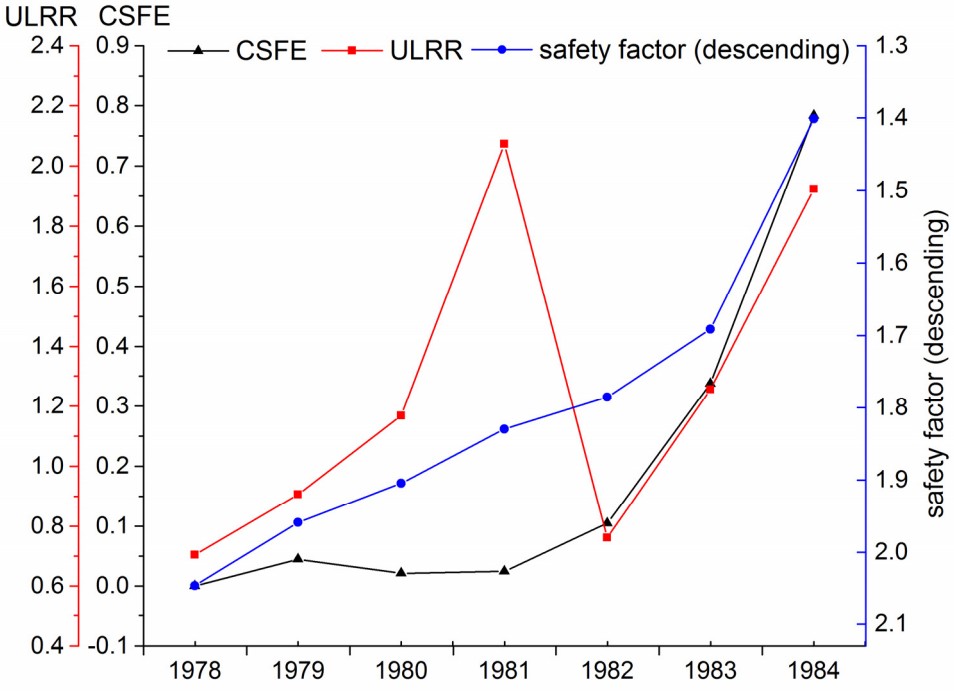

**Figure 8. Comparison of CSFE with ULRR and safety factor of Xintan landslide**

Similarly, state fusion entropy analysis of Baishuihe landslide, Bazimen landslide, Shuping landslide and Pajiayan landslide in the Three Gorges Reservoir area in China were also conducted and their results are shown in Figure 9. Similarities and differences between displacement and state fusion entropy are found through a comparative analysis of these landslides. As for Bazimen landslide and Pajiayan landslide, cumulative state fusion entropy and cumulative displacement show similar change rules especially during the drawdown period of water level, indicating their intrinsic consistency. As for Baishuihe landslide and Shuping landslide, cumulative state fusion entropy of shows a distinctly different characteristic from their cumulative displacement. Taking Baishuihe landslide as an example, the severe deformation in June 2007 seems to suggest that the landslide has entered accelerative deformation stage. However, subsequent monitoring has proved that the deformation is only a temporary effect of heavy rainfall and fluctuation of water level (Xu et al., 2008). In Figure 9, cumulative state fusion entropy of Baishuihe landslide returns to a low level after several historical maxima.

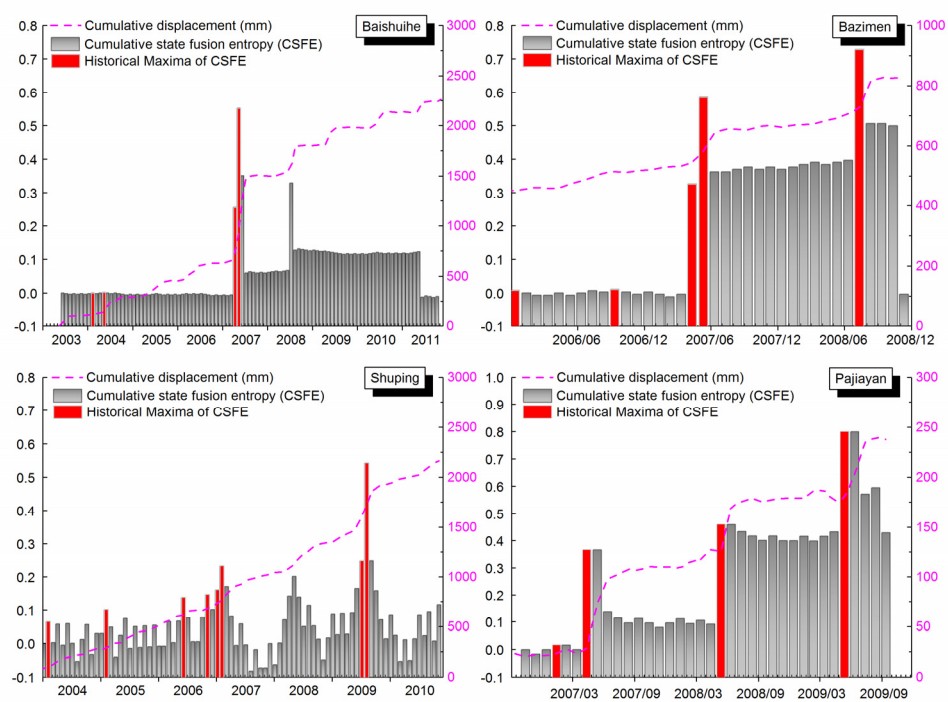

**Figure 9.** Cumulative state fusion entropy and historical maxima of Baishuihe, Bazimen, Shuping and Pajiayan landslide

## 4 Discussion and Conclusion

Under the guidance of dynamic state system and based on the relationship of displacement monitoring data, deformation state and landslide stability, a state fusion entropy approach is proposed to conduct a continuous and site-specific analysis of landslide stability changing regularities. A joint clustering method combining K-means and cloud model is firstly proposed to investigate landslide deformation states, and then a multi-attribute entropy analysis follows to estimate landslide instability. Furthermore, a historical maximum index is introduced for identifying key time nodes of stability changes. To verify the effectiveness of this approach, Xintan landslide is selected as a detailed case and four other landslides in the Three Gorges Reservoir area as brief cases. Taking Xintan landslide as an example, cumulative state fusion entropy mainly fluctuated around zero in the initial deformation stage and uniform deformation stage, but an obvious fluctuant increasing tendency appeared after Xintan landslide entered accelerative deformation stage. In the meanwhile, a thorough collection of the macroscopic proofs also suggests that historical maxima are highly consistent with landslide macroscopic deformation behaviours.

Compared with traditional safety factor, state fusion entropy evaluates the landslide instability, and is capable to indicate its extent and changing regularities. Compared with simulation methods for landslide stability analysis, this approach takes displacement monitoring data as the basis of landslide stability analysis, and thus is prone to continuous stability analysis. Compared with direct judgment from displacement monitoring data, this approach analyse landslide deformation states by a

data-driven model, avoiding the disunity of individual engineering geology experience, ensuring its applicability to the geological conditions of different landslides.

To measure the influence of different cluster numbers on the performance of this method, CSFE with different cluster numbers (K=3 to 7) is compared, as shown in Figure 10. As can be seen, CSFE varies slightly with K. This is mainly because that different K correspond to different division roughness of deformation states, which sequentially affects the value of the CSFE. Due to this fluctuation, CSFE is not intended to be applied to landslide early warning unless be qualified by further research. Despite this, the overall trend remains unchanged, which suggests a steady statistical regularity of deformation states.

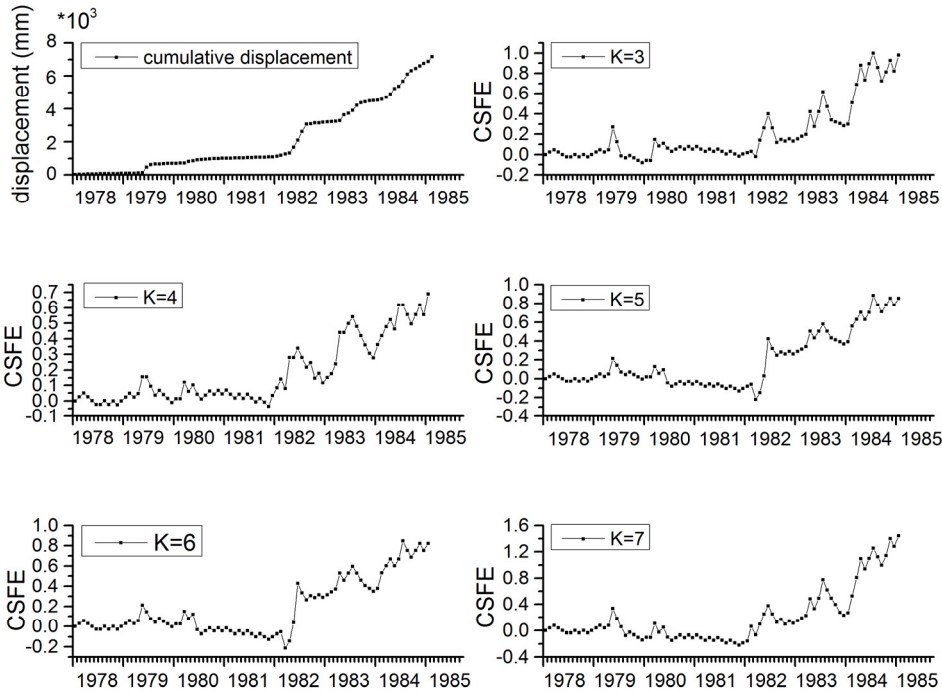

**Figure 10. CSFE with different cluster numbers (K=3 to 7) of Xintan landslide**

Furthermore, several issues also need to be clarified. Firstly, data selection and feature extraction are simplified. Multiple monitoring has already become a common practice in landslide monitoring, however the comprehensive mining of multi-source data is also still a common problem and relevant research of SFE is still in progress. Despite this, some thoughts have already emerged. For the open system of landslide, the displacement of different monitoring points can be regarded as landslide samples with different deformation scales on the one hand. On the other hand, fractal theory tells that same patterns as the entire system can be found if a small part of the whole be magnified. Therefore, fractal theory may contribute to multi-point data analysis. Besides, due to the lack of higher time resolution monitoring data, a common practice, selecting one typical monthly displacement data of GPS, is adopted for now. At this time scale, deformation velocity and acceleration are considered to express landslide deformation well and thus selected for deformation state definition. For higher time resolution data, some

feature extraction methods may be necessary to determine the *DE* and *DT* indexes. Finally, Entering into accelerative deformation stage is a necessary condition for landslide failure. Aiming at this, the fluctuant increasing tendency of cumulative state fusion entropy and the frequent renewal of historical maximum may help to judge whether landslide has entered accelerative deformation stage or not. Once this happens, other clues such as macro cracks should also be taken into account to fully determine landslide early warning level.

## Acknowledgements

This research was funded by the National Natural Sciences Foundation of China [grant numbers 41772376, 41302278]. The authors are grateful to the editors and reviewers for kind and constructive suggestions.

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
