# Peer review of "State fusion entropy for continuous and site-specific analysis of landslide stability changing regularities"

_Natural Hazards and Earth System Sciences, 2017_

## Referee Comment (RC1) · Anonymous Referee #1 · 23 Nov 2017

The paper presents a new data-driven methodology, based on a multi-attribute entropy analysis of deformation states which are obtained through joint clustering method combining K-means and cloud model. This method aims at identifying, at site-specific scale different state of activity of a landslide, in particular moment of acceleration or reduction of the displacement. The model was applied at different landslide test-case and it obtained consistent results respect to the real deformation patterns of the analyzed phenomena. The paper presents in details all the methodological approach and the achieved results. It represents an interesting model which could allow to improve the comprehension of the state of activity of slope instability, also in relation to an early-warning application. Instead, some aspects of the results presentation are incomplete, requiring clarifications and further explanations. Thus, several revisions are recom-

mended to improve the overall quality of the work. Suggested revisions follow:

General comments

In Introduction section, it is important to describe better the other methodologies indicated in the text (Saito's method, FEM, LEM), in particular their fundamental principles, the main advantages and limitations and their range of application. This can be reinforced further with references of significant works presented case studies of these applications.

In Introduction section, please indicate some works when displacement thresholds were defined and the values of these thresholds, in relation to the type of phenomenon and the geological context.

The developed methodology is a data-driven model, which is based on displacement data. For a better definition of the k-clusters, it could be necessary developing the method using real data where inactive, active, reactivated, and, also, failure states occurred during the considered measurement periods, as demonstrated in the analyzed case studies. Please, discuss about this aspect, in particular in relation to the potential ability of the methodology to identify the failure times of a landslide even if it has not been occurred yet.

Could this method be applicable also at higher time resolution of displacement data (e.g. daily, hourly)? This could improve the prediction for early warning applications. Please, insert a discussion about the aspect.

Please indicate if there are several references, in previous works, which highlight that the historical maxima identified by the model for each studied landslide are correspondent to acceleration/reactivation periods or failure moments.

Specific comments

pag. 2 line 11: The sentence is unclear. Please, clarify its concept, introducing other references, if it is necessary.

Please, substitute all the abbreviations in the text (e.g. 's, can't) with the corresponding entire terms. pag. 2 line 19: are there any previous works about entropy concepts application to landslides state of stability analysis? If yes, please refer to them and summarize their main achieved results.

In Methodology section: how many landslide deformation state can be identified by k-means/cloud analysis? This could have effects also on the definition of changing in landslide activity, e.g. a reactivation phase following a stable one.

Please, divide the description of the selected case studies from the results. Thus, it could be added a section ("Study area" or "Materials") before "Results" section.

It could be useful highlighting more geological and geomorphological features of both the study area and the test sites and also the triggering factors of the studied landslides.

"Discussion" and "Conclusion" section present several repetitions of the same concepts. It could be better merged these sections in another one ("Discussion and conclusion"), adding also references supporting the presented concepts.

Technical corrections

pag. 1 line 13: the evolutionary stages of the phenomenon

pag. 1 line 15: for assessing landslide stability

pag. 1 line 18: damages of properties every year

pag. 1 line: at site specific scale

pag. 2 line 5: it becomes of interest to find

pag. 2 line 9: Due to its easy acquisition

pag. 2 line 16: Previous works have introduced

pag. 3 line 8: the individuation of different deformations states

pag. 3 line 9: to investigate deformation states

pag. 7 line 18: As time goes on

pag. 8 line 4: with a length of 2000 m

pag. 8 line 17: monthly indexes for Xintan landslide

pag. 13 line 8: entering accelerative deformation stage highlighted in previous works (please insert references about this)

---

## Referee Comment (RC2) · Anonymous Referee #2 · 28 Nov 2017

The authors propose a new data-driven approach to quantify various states of activity of landslides and support, in perspective, decision-making within early-warning systems. The topic is undoubtedly interesting and with a potential of providing better information on site-specific landslide activity.

Nevertheless, I find that this paper does not really show whether the proposed approach gives a real advantage over other existing data-driven, empirical or physically-based methods in quantifying landslide stability/instability. A comparison of several methods would be greatly helpful.

Furthermore, there is no evidence that the method can be successfully used in an early-warning perspective, which is the goal set in the abstract. My main concern is that the entropy approach used by the authors is based solely on measurements of dis-

placements, seemingly in a single point of a landslide. The authors show that the pattern of state fusion entropy is (not surprisingly!) consistent with that of displacements (input information). Thus, what does the entropy tell in addition to what is already obvious by looking at the displacement pattern and, perhaps, by setting displacement rate thresholds to provide early warning? This has not been clarified. In addition, can the performance of the model be improved by integrating several displacement measurements (and perhaps pore pressures, water level, water content, deep deformations, etc.)? This is an important topic to be addressed.

It may be argued that the displacement rate thresholds are set arbitrarily in a displacement-based monitoring system. However, I see that even in this data-driven approach there are arbitrary site-specific decisions made by the authors (e.g. page 9 line 4), which perhaps can affect the model output. So, for a model to be truly data-driven, I expect no arbitrary choices, or arbitrary choices to have little influence: the dataset should provide the answer itself.

Finally, the content of the work does not seem to match its title: monthly displacements are probably too far from a "real-time" landslide monitoring when incipient failure is concerned. I expected to see interpretation of daily, hourly or even more frequent observations of landslide displacements prior to failure.

Due to these concerns, I feel that this manuscript is not ready for publication in the present form. I recommend the authors update their work by addressing the above points and, in particular, by including evidence of good performance of their model in making usable predictions of landslide failure based on high-resolution displacement patterns, which could be used in an early warning system.
* * *

---

## Referee Comment (RC3) · Anonymous Referee #3 · 29 Nov 2017

This paper proposes a new data-driven approach for real-time and site-specific analysis of landslide stability changing regularities based on a multi-attribute entropy analysis of deformation states from the aspect of landslide system. This approach was applied to different landslide and presented interesting results and could provide better information on site-specific landslide activity. Still, several revisions may help to improve the overall quality of the work. Firstly, the advantages and the limitations of existing methods seems too brief to emphasize the meaning and emergency of the proposed approach. The processes of the model is complex, please organize this part clearly. I suggest that the methods should be divided into several subsections. This method named "the proposed joint clustering method combining k-means and cloud model" should be refined. The part of "materials and results" should be correspon-

dence with the part of "methods". Secondly, in the "Deformation state definition based on K-means combined with Cloud Model", a better explanation why deformation rate and acceleration are selected to define deformation states may be necessary. How the displacement data was chosen because it is quite common for a landslide to have multiple displacement monitoring points at present. Thirdly, in the "materials and results" section, only monthly displacement data was used and it seems not very consistent with "real-time" in the title. Since for now monthly monitoring displacement is mainly adopted in most studies, "monthly stability" may be more appropriate for the title. In the meanwhile, the discussion on the process of other monitoring frequency data needs to be added. Finally, "Discussion" and "Conclusion" present several repetitions and need a better description. Meanwhile, the English written of this paper should be modified carefully again.

---

## Author Comment (AC1) · 12 Dec 2017

Dear Anonymous Referee #1:

First of all, we would like to express our sincere appreciation of your very constructive comments and suggestion.

Next, in a sequence, we would like to respond to comments in a point to point manner so that hopefully all the questions can be answered or clarified. All the answers and responses are in red.

The paper presents a new data-driven methodology, based on a multi-attribute entropy analysis of deformation states which are obtained through joint clustering method combining K-means and cloud model. This method aims at identifying, at site-specific scale different state of activity of a landslide, in particular moment of acceleration or reduction of the displacement. The model was applied at different landslide test-case and it obtained consistent results respect to the real deformation patterns of the analyzed phenomena. The paper presents in details all the methodological approach and the achieved results. It represents an interesting model which could allow to improve the comprehension of the state of activity of slope instability, also in relation to an early warning application.

Thanks for your encouraging words.

Instead, some aspects of the results presentation are incomplete, requiring clarifications and further explanations. Thus, several revisions are recommended to improve the overall quality of the work. Suggested revisions follow:

General comments

1) In Introduction section, it is important to describe better the other methodologies indicated in the text (Saito's method, FEM, LEM), in particular their fundamental principles, the main advantages and limitations and their range of application. This can be reinforced further with references of significant works presented case studies of these applications.

A more detailed introduction of other methodologies (Saito's method, FEM, LEM) has been added, including their fundamental principles, advantages and limitations.

Saito's method is an empirical forecast model and is suitable for the prediction of sliding tendency and then the failure time. Based on homogeneous soil creep theory and displacement curve, it divides displacement creep curves into three stages: deceleration creep, stable creep and accelerating creep, and establishes a differential equation for accelerating creep. The physical basis of Saito's method helped it to successfully forecast a landslide that occurred in Japan in December 1960, but also makes it strongly dependent on field observations. LEM is a kind of calculation method to evaluate landslide stability based on mechanical balance principle. By assuming a potential sliding surface and slicing the sliding body on the potential sliding surface firstly, LEM calculates the shear resistance and the shear force of each slice along the potential sliding surface and defines their ratio as the safety factor to describe landslide stability. LEM is simple and can directly analyse landslide stability under limit condition without geotechnical constitutive analysis. However, this neglect of geotechnical constitutive characteristic also restricts it to a static mechanics evaluation model that is incapable to evaluate the changing regularities of landslide stability. In the meanwhile, LEM involves too many physical parameters such as cohesive strength and friction angle, which

makes it greatly limited in landslide forecast and early warning. As a typical numerical simulation method, FEM subdivides a large problem into smaller, simpler parts that are called finite elements. The simple equations that model these finite elements are then assembled into a larger system of equations that models the entire problem. FEM then uses variational methods from the calculus of variations to approximate a solution by minimizing an associated error function. In landslide stability analysis, FEM can not only satisfy the static equilibrium condition and the geotechnical constitutive characteristic, but also adapt to the discontinuity and heterogeneity of the rock mass. However, FEM is quite sensitive to various involved parameters and the computation will increase greatly to get more accurate results. If parameters and boundaries are precisely determined, LEM and FEM can provide results with high reliability. [Has been added in "Introduction"]

2) In Introduction section, please indicate some works when displacement thresholds were defined and the values of these thresholds, in relation to the type of phenomenon and the geological context.

This paragraph has been rephrased and three references have been added.

Macciotta et al. (2016) suggested that velocity threshold be used as a criterion for early warning system and the annual horizontal displacement threshold for Ripley Landslide (GPS 1) can be 90 mm and that between May and September can be 25 mm. Based on the analysis of a large number of displacement monitoring data, Xu and Zeng (2009) proposed that deformation acceleration be used as an indicator of landslide warning, and the acceleration threshold of Jimingsi landslide was regarded as 0.45 mm/d2 and that of another landslide in Daye Iron Mine as 0.2 mm/d2. Federico et al. (2012) presented a systematic introduction to the prediction of landslide failure time according to the displacement data. [Has been added in "Introduction"]

3) The developed methodology is a data-driven model, which is based on displacement data. For a better definition of the k-clusters, it could be necessary developing the method using real data where inactive, active, reactivated, and, also, failure states occurred during the considered measurement periods, as demonstrated in the analyzed case studies. Please, discuss about this aspect, in particular in relation to the potential ability of the methodology to identify the failure times of a landslide even if it has not been occurred yet.

To verify the effectiveness of the state fusion entropy method, five landslides in the Three Gorges Reservoir area in China were selected as examples for stability changing regularities analysis. Among them, Xintan landslide is a reactive landslide triggered by rainfall and has failed. Baishuihe landslide, Bazimen landslide and Shuping landslide are reactive landslides mainly triggered by reservoir water level and rainfall. Pajiayan landslide is a new-born landslide. [Has been added in "Case study"]

For now, the state fusion entropy is designed without the function of forecasting but it still offers helps for landslide stability analysis and further the early warning. Cumulative state fusion entropy reflects the overall instability of landslide and its changing forms (fluctuation around zero type and fluctuant increasing type) also do help to judge landslide evolutionary stages and deformation tendency. Besides, the historical maximum index indicates the renewal of the most dangerous state of the landslide and may server as a new clue for landslide early warning. But this new clue should not be exaggerated to such an extent that other clues can all be replaced. Once historical maximum is renewed frequently, other clues such as macro cracks should also be taken into account to fully determine landslide early warning level. [Has been

4)  Could this method be applicable also at higher time resolution of displacement data (e.g. daily, hourly)? This could improve the prediction for early warning applications. Please, insert a discussion about the aspect.

    While defining deformation states, deformation velocity and acceleration are selected because they are considered to represent the landslide deformation characteristics well on the assumption that displacement is monitored monthly. At this time scale, the monitoring error of GPS can be ignored compared to landslide actual deformation. However, as the time resolution of displacement monitoring data increases, the impact of monitoring errors will be greater. In this case, landslide deformation features may not be deformation velocity and acceleration but determined by some feature extraction methods. Neglecting the consideration of monitoring error, the method is capable to monitoring data with higher time resolution and corresponding feature extraction methods are under study.

5)  Please indicate if there are several references, in previous works, which highlight that the historical maxima identified by the model for each studied landslide are correspondent to acceleration/reactivation periods or failure moments.

    More macroscopic phenomenon has been added as the evidence to validate the effectiveness of this method.

    The macroscopic behaviors of Xintan landslide near historical maxima was investigated according to previous studies (Wang, 1996). In June 1982, some trees in the top area of Jiangjiapo were dumped. A small amount of north-west tensile cracks appeared on the steeper section of the east. Around August 1982, the front edge of Jiangjiapo went through a small collapse. In June 1983, the colluvial deposits between Guangjiaya and Jiangjiapo showed signs of resurrection. At the end of 1984, the trailing edge of the landslide showed an "armchair" shape and the leading edge was bulged out. Some collapse pits were found on the upper side while several new tensile cracks in the middle. Meanwhile, some small collapses which seem irrelevant to rainfall occurred. In May 1985, old cracks widened and new cracks appeared, forming a ladder-shaped landing ridge. Moreover, Jiangjiapo presented a clear trend of the overall slippage. These proofs suggest that the historical maximum index is highly consistent with landslide macroscopic deformation behaviors. [Has been added in "Introduction"]

Specific comments

1)  Page. 2 line 11: The sentence is unclear. Please, clarify its concept, introducing other references, if it is necessary.

    The sentence has been rephrased.

2)  Please, substitute all the abbreviations in the text (e.g. 's, can't) with the corresponding entire terms.

    Abbreviations like "it's" and "can't" has been substituted. Saito's method is reserved because it is the name of one method.

3)  Page. 2 line 19: are there any previous works about entropy concepts application to landslides state of stability analysis? If yes, please refer to them and summarize their main achieved results.

    At present, studies about entropy concepts application to landslides state of stability analysis are quite rare. Nevertheless, we detailed one literature and summarized its results and

disadvantages.

Shi and Jin (2009) proposed a generalized information entropy approach (GIE) to evaluate the "energy" of multi-triggers of landslide and found that the GIE index showed a mutation before landslide failure in a case study. But this GIE method is aimed at landslide triggering factors and thus cannot directly indicate landslide stability because of the ignorance of energy transfer efficiency between triggering factors and landslide. [Has been added in "Introduction"]

4) In Methodology section: how many landslide deformation state can be identified by k-means/cloud analysis? This could have effects also on the definition of changing in landslide activity, e.g. a reactivation phase following a stable one.

Theoretically, the k-means clustering method is based on the data distribution of input data. The cluster number K only determines the division roughness of clusters and has little impact on the distribution of clusters which is the basis of the state fusion entropy approach. Therefore, the cluster number was empirically set to 3 in the case study. Now some strategies have been proposed to determine cluster number totally and automatically according to input data. And this can also be used as an improvement of the method.

5) Please, divide the description of the selected case studies from the results. Thus, it could be added a section ("Study area" or "Materials") before "Results" section.

Thanks for your suggestion. This paper mainly analyses the landslide stability changing regularities from the perspective of landslide system entropy. Five landslides in the Three Gorges Reservoir area are selected in which Xintan landslide is selected as a detailed case study and other four landslides as brief ones. If divide the description of the selected case studies from the results, a detailed description of the background information (including geological and geomorphological features, triggers, etc.) of all these five landslide will be required, which may take too much space. Meanwhile, the part of the deformation states definition may be too thin after this division. Thanks for your kind suggestion, and we may choose "Case Study" for the section.

6) It could be useful highlighting more geological and geomorphological features of both the study area and the test sites and also the triggering factors of the studied landslides.

Explained in the former comment.

7) "Discussion" and "Conclusion" section present several repetitions of the same concepts. It could be better merged these sections in another one ("Discussion and conclusion"), adding also references supporting the presented concepts.

Thanks for your constructive suggestion. We have merged and rephrased the "Discussion" and "Conclusion". The revised "Discussion and conclusion" section is as follows.

Under the guidance of dynamic state system and based on the relationship of displacement monitoring data, deformation state and landslide stability, a state fusion entropy approach is proposed to conduct a real-time and site-specific analysis of landslide stability changing regularities. A joint clustering method combining K-means and cloud model is firstly proposed to investigate landslide deformation states, and then a multi-attribute entropy analysis follows to estimate landslide instability. Furthermore, a historical maximum index is introduced for landslide early warning. To verify the effectiveness of this approach, Xintan landslide is selected as a detailed case and four other landslides in the Three Gorges Reservoir area as brief cases. Taking Xintan landslide as an example, cumulative state fusion entropy mainly fluctuated around zero in the initial deformation stage and uniform deformation stage, but an

obvious fluctuant increasing tendency appeared after Xintan landslide entered accelerative deformation stage. In the meanwhile, a thorough collection of the macroscopic proofs also suggested that historical maxima are highly consistent with landslide macroscopic deformation behaviors.

Compared with traditional safety factor, state fusion entropy evaluates the landslide instability, and is capable to indicate its extent and changing regularities. Compared with simulation methods for landslide stability analysis, this approach takes displacement monitoring data as the basis of landslide stability analysis, and thus is prone to real-time stability analysis. Compared with direct judgment from deformation velocity and acceleration, this approach analyse landslide deformation states by a data-driven model, avoiding the disunity of individual engineering geology experience, ensuring its applicability to the geological conditions of different landslides.

However, several issues also need to be clarified. The landslide stability changing regularities are obtained by comparing current stability with the past stability and thus it is meaningless to compare the state fusion entropy of different landslides. In addition, if displacement monitoring data only covers one evolutionary stage, cumulative state fusion entropy may not present the fluctuant increasing trend but a relatively simple curve with only a few historical maxima. For now, the state fusion entropy is designed without the function of forecasting but it still offers helps for landslide stability analysis and further the early warning. Cumulative state fusion entropy reflects the overall instability of landslide and its changing forms (fluctuation around zero type and fluctuant increasing type) also do help to judge landslide evolutionary stages and deformation tendency. Besides, the historical maximum index indicates the renewal of the most dangerous state of the landslide and may server as a new clue for landslide early warning. But this new clue should not be exaggerated to such an extent that other clues can all be replaced. Once historical maximum is renewed frequently, other clues such as macro cracks should also be taken into account to fully determine landslide early warning level.

Technical corrections
1. pag. 1 line 13: the evolutionary stages of the phenomenon Modified
2. pag. 1 line 15: for assessing landslide stability Modified
3. pag. 1 line 18: damages of properties every year Modified
4. pag. 1 line: at site specific scale Modified
5. pag. 2 line 5: it becomes of interest to find Modified
6. pag. 2 line 9: Due to its easy acquisition Modified
7. pag. 2 line 16: Previous works have introduced Modified
8. pag. 3 line 8: the individuation of different deformations states Modified
9. pag. 3 line 9: to investigate deformation states Modified
10. pag. 7 line 18: As time goes on Modified
11. pag. 8 line 4: with a length of 2000 m Modified
12. pag. 8 line 17: monthly indexes for Xintan landslide Modified
13. pag. 13 line 8: entering accelerative deformation stage highlighted in previous works (please insert references about this) Modified

---

## Author Comment (AC2) · 12 Dec 2017

Dear Anonymous Referee #2:

First of all, we would like to express our sincere appreciation of your very constructive comments and suggestion.

Next, in a sequence, we would like to respond to comments in a point to point manner so that hopefully all the questions can be answered or clarified. All the answers and responses are in red.

The authors propose a new data-driven approach to quantify various states of activity of landslides and support, in perspective, decision-making within early-warning systems. The topic is undoubtedly interesting and with a potential of providing better information on site-specific landslide activity.

Thanks very much for your encouraging words.

Nevertheless, I find that this paper does not really show whether the proposed approach gives a real advantage over other existing data-driven, empirical or physically based methods in quantifying landslide stability/instability. A comparison of several methods would be greatly helpful.

In introduction, the basic principles and main advantages and disadvantages of the existing methods (Saito's method, FEM, LEM) are detailed, hoping to highlight the starting point of this paper. In the case study, it is difficult to compare the results of this method with other methods whose results usually are presented with safety factor, because this paper indicates landslide instability with the proposed state fusion entropy index. As a supplement, more macroscopic phenomenon has been added as the evidence to validate the effectiveness of this method.

Saito's method is an empirical forecast model and is suitable for the prediction of sliding tendency and then the failure time. Based on homogeneous soil creep theory and displacement curve, it divides displacement creep curves into three stages: deceleration creep, stable creep and accelerating creep, and establishes a differential equation for accelerating creep. The physical basis of Saito's method helped it to successfully forecast a landslide that occurred in Japan in December 1960, but also makes it strongly dependent on field observations. LEM is a kind of calculation method to evaluate landslide stability based on mechanical balance principle. By assuming a potential sliding surface and slicing the sliding body on the potential sliding surface firstly, LEM calculates the shear resistance and the shear force of each slice along the potential sliding surface and defines their ratio as the safety factor to describe landslide stability. LEM is simple and can directly analyse landslide stability under limit condition without geotechnical constitutive analysis. However, this neglect of geotechnical constitutive characteristic also restricts it to a static mechanics evaluation model that is incapable to evaluate the changing regularities of landslide stability. In the meanwhile, LEM involves too many physical parameters such as cohesive strength and friction angle, which makes it greatly limited in landslide forecast and early warning. As a typical numerical simulation method, FEM subdivides a large problem into smaller, simpler parts that are called finite elements. The simple equations that model these finite elements are then assembled into a larger system of equations that models the entire problem. FEM then uses variational methods from the calculus of variations to approximate a solution by minimizing an associated error function. In landslide stability analysis, FEM can not only satisfy the static equilibrium condition and the geotechnical constitutive characteristic, but also adapt to the discontinuity and heterogeneity of the rock mass. However, FEM is quite sensitive to various involved parameters and the computation will increase

greatly to get more accurate results. If parameters and boundaries are precisely determined, LEM and FEM can provide results with high reliability. [Has been added in "Introduction"]

The macroscopic behaviors of Xintan landslide near historical maxima was investigated according to previous studies (Wang, 1996). In June 1982, some trees in the top area of Jiangjiapo were dumped. A small amount of north-west tensile cracks appeared on the steeper section of the east. Around August 1982, the front edge of Jiangjiapo went through a small collapse. In June 1983, the colluvial deposits between Guangjiaya and Jiangjiapo showed signs of resurrection. At the end of 1984, the trailing edge of the landslide showed an "armchair" shape and the leading edge was bulged out. Some collapse pits were found on the upper side while several new tensile cracks in the middle. Meanwhile, some small collapses which seem irrelevant to rainfall occurred. In May 1985, old cracks widened and new cracks appeared, forming a ladder-shaped landing ridge. Moreover, Jiangjiapo presented a clear trend of the overall slippage. These proofs suggest that the historical maximum index is highly consistent with landslide macroscopic deformation behaviors. [Has been added in "Introduction"]

Furthermore, there is no evidence that the method can be successfully used in an early-warning perspective, which is the goal set in the abstract. My main concern is that the entropy approach used by the authors is based solely on measurements of displacements, seemingly in a single point of a landslide. The authors show that the pattern of state fusion entropy is (not surprisingly!) consistent with that of displacements (input information). Thus, what does the entropy tell in addition to what is already obvious by looking at the displacement pattern and, perhaps, by setting displacement rate thresholds to provide early warning? This has not been clarified. In addition, can the performance of the model be improved by integrating several displacement measurements (and perhaps pore pressures, water level, water content, deep deformations, etc.)? This is an important topic to be addressed.

Thanks for your comments. This approach is proposed to analyse landslide stability changing regularities and further provide clues for landslide early warning. The cumulative state fusion entropy may be similar to cumulative displacement (Xintan landslide). However, they can also be very different which has been presented in the result interpretation of Baishuihe landslide. As for the data selection. Nowadays, one landslide may be monitored by multiple monitors with multiple sensors and various data can be obtained such as surface displacement, deep displacement, pore pressure, water content and so on. There is no doubt that all these monitoring data contain the information about landslide state and much more comprehensive landslide states can be obtained if all these monitoring data are utilized. However, this comprehensive monitoring data is not yet common. And thus a traditional operation, selecting one typical displacement data of GPS, is adopted for generality and simplicity. Research of multi-monitoring and multi-sensor data fusion has been carried.

It may be argued that the displacement rate thresholds are set arbitrarily in a displacement-based monitoring system. However, I see that even in this data-driven approach there are arbitrary site-specific decisions made by the authors (e.g. page 9 line 4), which perhaps can affect the model output. So, for a model to be truly data driven, I expect no arbitrary choices, or arbitrary choices to have little influence: the dataset should provide the answer itself.

Thanks for your advice. Theoretically, the k-means clustering method is based on the data distribution of input data. The cluster number K only determines the division roughness of clusters

and has little impact on the distribution of clusters which is the basis of the state fusion entropy approach. Therefore, the cluster number was empirically set to 3 in the case study. Now some strategies have been proposed to determine cluster number totally and automatically according to input data. And this can also be used as an improvement of the method.

Finally, the content of the work does not seem to match its title: monthly displacements are probably too far from a "real-time" landslide monitoring when incipient failure is concerned. I expected to see interpretation of daily, hourly or even more frequent observations of landslide displacements prior to failure.

Thanks for your constructive suggestion. While defining deformation states, deformation velocity and acceleration are selected because they are considered to represent the landslide deformation characteristics well on the assumption that displacement is monitored monthly. At this time scale, the monitoring error of GPS can be ignored compared to landslide actual deformation. However, as the time resolution of displacement monitoring data increases, the impact of monitoring errors will be greater. In this case, landslide deformation features may not be deformation velocity and acceleration but determined by some feature extraction methods. Neglecting the consideration of monitoring error, the method is capable to monitoring data with higher time resolution and corresponding feature extraction methods are under study.

Due to these concerns, I feel that this manuscript is not ready for publication in the present form. I recommend the authors update their work by addressing the above points and, in particular, by including evidence of good performance of their model in making usable predictions of landslide failure based on high-resolution displacement patterns, which could be used in an early warning system.

For now, the state fusion entropy is designed without the function of forecasting but it still offers helps for landslide stability analysis and further the early warning. Cumulative state fusion entropy reflects the overall instability of landslide and its changing forms (fluctuation around zero type and fluctuant increasing type) also do help to judge landslide evolutionary stages and deformation tendency. Besides, the historical maximum index indicates the renewal of the most dangerous state of the landslide and may server as a new clue for landslide early warning. But this new clue should not be exaggerated to such an extent that other clues can all be replaced. Once historical maximum is renewed frequently, other clues such as macro cracks should also be taken into account to fully determine landslide early warning level. [Has been added in "Discussion and conclusion"]

---

## Author Comment (AC3) · 12 Dec 2017

Dear Anonymous Referee #3:

First of all, we would like to express our sincere appreciation of your very constructive comments and suggestion.

Next, in a sequence, we would like to respond to comments in a point to point manner so that hopefully all the questions can be answered or clarified. All the answers and responses are in red.

This paper proposes a new data-driven approach for real-time and site-specific analysis of landslide stability changing regularities based on a multi-attribute entropy analysis of deformation states from the aspect of landslide system. This approach was applied to different landslide and presented interesting results and could provide better information on site-specific landslide activity.

Thanks for your encouraging words.

Still, several revisions may help to improve the overall quality of the work.

Firstly, the advantages and the limitations of existing methods seems too brief to emphasize the meaning and emergency of the proposed approach. The processes of the model is complex, please organize this part clearly. I suggest that the methods should be divided into several subsections. This method named "the proposed joint clustering method combining k-means and cloud model" should be refined. The part of "materials and results" should be correspondence with the part of "methods".

Thanks for your kind suggestion. Firstly, a detailed introduction to these methods (Saito's method, LEM and FEM) has been added, including their advantages and the limitations. Given that several methods are involved in this approach, we have tried our best to divide it into two near-independent parts, respectively the definition and the multi-attribute entropy analysis of deformation states. Too much sub-section may undermine the integrity of the content. The method name "the proposed joint clustering method combining k-means and cloud model" may be too long but it expresses apparently the essential factors of this method. K-means and cloud model complement each other, together form the core of the joint clustering. Sorry, we have not figure out a better alternative. Any suggestions and advices on this issue are always welcome.

Saito's method is an empirical forecast model and is suitable for the prediction of sliding tendency and then the failure time. Based on homogeneous soil creep theory and displacement curve, it divides displacement creep curves into three stages: deceleration creep, stable creep and accelerating creep, and establishes a differential equation for accelerating creep. The physical basis of Saito's method helped it to successfully forecast a landslide that occurred in Japan in December 1960, but also makes it strongly dependent on field observations. LEM is a kind of calculation method to evaluate landslide stability based on mechanical balance principle. By assuming a potential sliding surface and slicing the sliding body on the potential sliding surface firstly, LEM calculates the shear resistance and the shear force of each slice along the potential sliding surface and defines their ratio as the safety factor to describe landslide stability. LEM is simple and can directly analyse landslide stability under limit condition without geotechnical constitutive analysis. However, this neglect of geotechnical constitutive characteristic also restricts it to a static mechanics evaluation model that is incapable to evaluate the changing regularities of landslide stability. In the meanwhile, LEM involves too many physical parameters such as cohesive strength and friction angle, which makes it greatly limited in landslide forecast and early warning. As a typical numerical simulation method, FEM subdivides a large problem into smaller, simpler parts that are called finite elements. The

simple equations that model these finite elements are then assembled into a larger system of equations that models the entire problem. FEM then uses variational methods from the calculus of variations to approximate a solution by minimizing an associated error function. In landslide stability analysis, FEM can not only satisfy the static equilibrium condition and the geotechnical constitutive characteristic, but also adapt to the discontinuity and heterogeneity of the rock mass. However, FEM is quite sensitive to various involved parameters and the computation will increase greatly to get more accurate results. If parameters and boundaries are precisely determined, LEM and FEM can provide results with high reliability. [Has been added in "Introduction"]

Secondly, in the "Deformation state definition based on K-means combined with Cloud Model", a better explanation why deformation rate and acceleration are selected to define deformation states may be necessary. How the displacement data was chosen because it is quite common for a landslide to have multiple displacement monitoring points at present.

Thanks for your kind advice.

1) Why we only select one typical displacement data:

    Nowadays, one landslide may be monitored by multiple monitors with multiple sensors and various data can be obtained such as surface displacement, deep displacement, pore pressure, water content and so on. There is no doubt that all these monitoring data contain the information about landslide state and much more comprehensive landslide state can be obtained if all these monitoring data are utilized. However, this comprehensive monitoring data is not yet common. And thus a traditional operation, selecting one typical displacement data of GPS, is adopted for generality and simplicity. Research of multi-monitoring and multi-sensor data fusion has been carried.

2) why deformation rate and acceleration are selected to define deformation states

    The essence of this problem is how to determine the deformation features of displacement monitoring data. While defining deformation states, deformation velocity and acceleration are selected because they are considered to represent the landslide deformation characteristics well on the assumption that displacement is monitored monthly. At this time scale, the monitoring error of GPS can be ignored compared to landslide actual deformation. However, as the time resolution of displacement monitoring data increases, the impact of monitoring errors will be greater. In this case, landslide deformation features may not be deformation velocity and acceleration but determined by some feature extraction methods. Neglecting the consideration of monitoring error, the method is capable to monitoring data with higher time resolution and corresponding feature extraction methods are under study.

Thirdly, in the "materials and results" section, only monthly displacement data was used and it seems not very consistent with "real-time" in the title. Since for now monthly monitoring displacement is mainly adopted in most studies, "monthly stability" may be more appropriate for the title. In the meanwhile, the discussion on the process of other monitoring frequency data needs to be added.

Thanks for your kind suggestion. The doubt about data selection has been explained in the former question. Simply speaking, this approach is capable to monitoring data with high time solution. But for generality and simplicity, monthly monitoring data is selected in this paper on the consideration that it is the most adopted data for now.

Finally, "Discussion" and "Conclusion" present several repetitions and need a better description. Meanwhile, the English written of this paper should be modified carefully again.

Thanks for your constructive suggestion. We have merged and rephrased the "Discussion" and "Conclusion". The revised "Discussion and conclusion" section is as follows.

Under the guidance of dynamic state system and based on the relationship of displacement monitoring data, deformation state and landslide stability, a state fusion entropy approach is proposed to conduct a real-time and site-specific analysis of landslide stability changing regularities. A joint clustering method combining K-means and cloud model is firstly proposed to investigate landslide deformation states, and then a multi-attribute entropy analysis follows to estimate landslide instability. Furthermore, a historical maximum index is introduced for landslide early warning. To verify the effectiveness of this approach, Xintan landslide is selected as a detailed case and four other landslides in the Three Gorges Reservoir area as brief cases. Taking Xintan landslide as an example, cumulative state fusion entropy mainly fluctuated around zero in the initial deformation stage and uniform deformation stage, but an obvious fluctuant increasing tendency appeared after Xintan landslide entered accelerative deformation stage. In the meanwhile, a thorough collection of the macroscopic proofs also suggested that historical maxima are highly consistent with landslide macroscopic deformation behaviors.

Compared with traditional safety factor, state fusion entropy evaluates the landslide instability, and is capable to indicate its extent and changing regularities. Compared with simulation methods for landslide stability analysis, this approach takes displacement monitoring data as the basis of landslide stability analysis, and thus is prone to real-time stability analysis. Compared with direct judgment from deformation velocity and acceleration, this approach analyse landslide deformation states by a data-driven model, avoiding the disunity of individual engineering geology experience, ensuring its applicability to the geological conditions of different landslides.

However, several issues also need to be clarified. The landslide stability changing regularities are obtained by comparing current stability with the past stability and thus it is meaningless to compare the state fusion entropy of different landslides. In addition, if displacement monitoring data only covers one evolutionary stage, cumulative state fusion entropy may not present the fluctuant increasing trend but a relatively simple curve with only a few historical maxima. For now, the state fusion entropy is designed without the function of forecasting but it still offers helps for landslide stability analysis and further the early warning. Cumulative state fusion entropy reflects the overall instability of landslide and its changing forms (fluctuation around zero type and fluctuant increasing type) also do help to judge landslide evolutionary stages and deformation tendency. Besides, the historical maximum index indicates the renewal of the most dangerous state of the landslide and may server as a new clue for landslide early warning. But this new clue should not be exaggerated to such an extent that other clues can all be replaced. Once historical maximum is renewed frequently, other clues such as macro cracks should also be taken into account to fully determine landslide early warning level.

---

## Author Comment (AC6) · 12 Dec 2017

The comment was uploaded in the form of a supplement:
https://www.nat-hazards-earth-syst-sci-discuss.net/nhess-2017-351/nhess-2017-351-AC6-supplement.pdf
* * *

---

## Author Response (AR1)

**The point to point response to the reviewers**

First of all, we would like to express our sincere appreciation of your very constructive comments and suggestion.

Next, in a sequence, we would like to respond to comments in a point to point manner so that hopefully all the questions can be answered or clarified. All the responses are in red and all the changes made in the manuscript are underlined.

5      **Response to Anonymous Referee #1:**

General comments

1.  In Introduction section, it is important to describe better the other methodologies indicated in the text (Saito's method, FEM, LEM), in particular their fundamental principles, the main advantages and limitations and their range of application. This can be reinforced further with references of significant works presented case studies of these applications.

10  A more detailed introduction of other methodologies (Saito's method, FEM, LEM) has been added, including their fundamental principles, advantages and limitations.

Saito's method is an empirical forecast model and is suitable for the prediction of sliding tendency and then the failure time. Based on homogeneous soil creep theory and displacement curve, it divides displacement creep curves into three stages: deceleration creep, stable creep and accelerating creep, and establishes a differential equation for accelerating creep. The

15  physical basis of Saito's method helped it to successfully forecast a landslide that occurred in Japan in December 1960, but also makes it strongly dependent on field observations. LEM is a kind of calculation method to evaluate landslide stability based on mechanical balance principle. By assuming a potential sliding surface and slicing the sliding body on the potential sliding surface firstly, LEM calculates the shear resistance and the shear force of each slice along the potential sliding surface and defines their ratio as the safety factor to describe landslide stability. LEM is simple and can directly analyse landslide

20  stability under limit condition without geotechnical constitutive analysis. However, this neglect of geotechnical constitutive characteristic also restricts it to a static mechanics evaluation model that is incapable to evaluate the changing regularities of landslide stability. In the meanwhile, LEM involves too many physical parameters such as cohesive strength and friction angle, which makes it greatly limited in landslide forecast and early warning. As a typical numerical simulation method, FEM subdivides a large problem into smaller, simpler parts that are called finite elements. The simple equations that model these

25  finite elements are then assembled into a larger system of equations that models the entire problem. FEM then uses variational methods from the calculus of variations to approximate a solution by minimizing an associated error function. In landslide stability analysis, FEM can not only satisfy the static equilibrium condition and the geotechnical constitutive characteristic, but also adapt to the discontinuity and heterogeneity of the rock mass. However, FEM is quite sensitive to various involved parameters and the computation will increase greatly to get more accurate results. If parameters and boundaries are precisely

30  determined, LEM and FEM can provide results with high reliability. [Has been added in "Introduction"]

2. In Introduction section, please indicate some works when displacement thresholds were defined and the values of these thresholds, in relation to the type of phenomenon and the geological context.

This paragraph has been rephrased and three references have been added.

Macciotta et al. (2016) suggested that velocity threshold be used as a criterion for early warning system and the annual horizontal displacement threshold for Ripley Landslide (GPS 1) can be 90 mm and that between May and September can be 25 mm. Based on the analysis of a large number of displacement monitoring data, Xu and Zeng (2009) proposed that deformation acceleration be used as an indicator of landslide warning, and the acceleration threshold of Jimingsi landslide was regarded as 0.45 mm/d$^2$ and that of another landslide in Daye Iron Mine as 0.2 mm/d$^2$. Federico et al. (2012) presented a systematic introduction to the prediction of landslide failure time according to the displacement data. [Has been added in "Introduction"]

3. The developed methodology is a data-driven model, which is based on displacement data. For a better definition of the k-clusters, it could be necessary developing the method using real data where inactive, active, reactivated, and, also, failure states occurred during the considered measurement periods, as demonstrated in the analysed case studies. Please, discuss about this aspect, in particular in relation to the potential ability of the methodology to identify the failure times of a landslide even if it has not been occurred yet.

To verify the effectiveness of the state fusion entropy method, five landslides in the Three Gorges Reservoir area in China were selected as examples for stability changing regularities analysis. Among them, Xintan landslide is a reactive landslide triggered by rainfall and has failed. Baishuihe landslide, Bazimen landslide and Shuping landslide are reactive landslides mainly triggered by reservoir water level and rainfall. Pajiayan landslide is a new-born landslide. [Has been added in "Case study"]

Entering into accelerative deformation stage is a necessary condition for landslide failure. Aiming at this, the fluctuant increasing tendency of cumulative state fusion entropy and the frequent renewal of historical maximum may help to judge whether landslide has entered accelerative deformation stage or not. Once this happens, other clues such as macro cracks should also be taken into account to fully determine landslide early warning level. In addition, the Markov property of deformation state can be used for prediction. [Has been added in "Discussion and conclusion"]

4. Could this method be applicable also at higher time resolution of displacement data (e.g. daily, hourly)? This could improve the prediction for early warning applications. Please, insert a discussion about the aspect.

The description of the inputs of the joint clustering method has been modified to avoid confusion with the ($v$, $a$) in case study. Moreover, the reason why deformation velocity and acceleration are selected in case study has been addressed. This method is capable for higher time resolution data, but the data features may need to be determined by some feature extraction methods. To describe clearly this method, two functional data type are defined for landslide displacement data. One is to indicate deformation extent (*DE*) and the other to deformation tendency (*DT*). Positive *DT* indicates an increasing deformation. The process of defining deformation states is as follows. Step 0. Unite *DE* and *DT* at the same time as an item, i.e., (*DE*, *DT*); [Has been added in 2.1]

Considering that the monitoring error of GPS can be ignored compared to landslide actual deformation on monthly time scale, monthly deformation velocity ($v$) was selected as the *DE* index and monthly deformation acceleration ($a$) as the *DT* index. [Has been added in Case study]

Besides, displacement data is currently obtained monthly by GPS. At this time scale, deformation velocity and acceleration are considered to express landslide deformation well and thus selected for deformation state definition. For higher time resolution data, some feature extraction methods may be necessary to determine the *DE* and *DT* indexes. [Has been added in "Conclusion and Discussion"]

5. Please indicate if there are several references, in previous works, which highlight that the historical maxima identified by the model for each studied landslide are correspondent to acceleration/reactivation periods or failure moments.

More macroscopic phenomenon has been added as the evidence to validate the effectiveness of this method.

The macroscopic behaviours of Xintan landslide near historical maxima were investigated according to previous studies (Wang, 1996). In June 1982, some trees in the top area of Jiangjiapo were dumped. A small amount of north-west tensile cracks appeared on the steeper section of the east. Around August 1982, the front edge of Jiangjiapo went through a small collapse. In June 1983, the colluvial deposits between Guangjiaya and Jiangjiapo showed signs of resurrection. At the end of 1984, the trailing edge of the landslide showed an "armchair" shape and the leading edge was bulged out. Some collapse pits were found on the upper side while several new tensile cracks in the middle. Meanwhile, some small collapses which seem irrelevant to rainfall occurred. In May 1985, old cracks widened and new cracks appeared, forming a ladder-shaped landing ridge. Moreover, Jiangjiapo presented a clear trend of the overall slippage. These proofs suggest that the historical maximum index is highly consistent with landslide macroscopic deformation behaviours. [Has been added in "Case study"]

**Specific comments**

1. Page. 2 line 11: The sentence is unclear. Please, clarify its concept, introducing other references, if it is necessary.

The sentence has been rephrased.

2. Please, substitute all the abbreviations in the text (e.g. 's, can't) with the corresponding entire terms.

Abbreviations like "it's" and "can't" has been substituted. Saito's method is reserved because it is the name of one method.

3. Page. 2 line 19: are there any previous works about entropy concepts application to landslides state of stability analysis? If yes, please refer to them and summarize their main achieved results.

At present, studies about entropy concepts application to landslides state of stability analysis are quite rare. Nevertheless, we detailed one literature and summarized its results and disadvantages.

Shi and Jin (2009) proposed a generalized information entropy approach (GIE) to evaluate the "energy" of multi-triggers of landslide and found that the GIE index showed a mutation before landslide failure in a case study. But this GIE method is aimed at landslide triggering factors and thus cannot directly indicate landslide stability because of the ignorance of energy transfer efficiency between triggering factors and landslide. [Has been added in "Introduction"]

4. In Methodology section: how many landslide deformation state can be identified by k-means/cloud analysis? This could have effects also on the definition of changing in landslide activity, e.g. a reactivation phase following a stable one.

Theoretically, the k-means clustering method is based on the data distribution of input data. The cluster number K only determines the division roughness of clusters and has little impact on the distribution of clusters which is the basis of the state fusion entropy approach. Therefore, the cluster number was empirically set to 3 in the case study. Now some strategies have been proposed to determine cluster number totally and automatically according to input data. And this can also be used as an improvement of the method.

5. Please, divide the description of the selected case studies from the results. Thus, it could be added a section ("Study area" or "Materials") before "Results" section.

Thanks for your suggestion. This paper mainly analyses the landslide stability changing regularities from the perspective of landslide system entropy. Five landslides in the Three Gorges Reservoir area are selected in which Xintan landslide is selected as a detailed case study and other four landslides as brief ones. If divide the description of the selected case studies from the results, a detailed description of the background information (including geological and geomorphological features, triggers, etc.) of all these five landslide will be required, which may take too much space. Meanwhile, the part of the deformation states definition may be too thin after this division. Thanks for your kind suggestion, and we may choose "Case Study" for the section.

6. It could be useful highlighting more geological and geomorphological features of both the study area and the test sites and also the triggering factors of the studied landslides.

This has been explained in the response to the former comment (Specific comment 5).

7. "Discussion" and "Conclusion" section present several repetitions of the same concepts. It could be better merged these sections in another one ("Discussion and conclusion"), adding also references supporting the presented concepts.

Thanks for your constructive suggestion. We have merged and rephrased the "Discussion" and "Conclusion". The revised "Discussion and conclusion" section is as follows.

Under the guidance of dynamic state system and based on the relationship of displacement monitoring data, deformation state and landslide stability, a state fusion entropy approach is proposed to conduct a continuous and site-specific analysis of landslide stability changing regularities. A joint clustering method combining K-means and cloud model is firstly proposed to investigate landslide deformation states, and then a multi-attribute entropy analysis follows to estimate landslide instability. Furthermore, a historical maximum index is introduced for landslide early warning. To verify the effectiveness of this approach, Xintan landslide is selected as a detailed case and four other landslides in the Three Gorges Reservoir area as brief cases. Taking Xintan landslide as an example, cumulative state fusion entropy mainly fluctuated around zero in the initial deformation stage and uniform deformation stage, but an obvious fluctuant increasing tendency appeared after Xintan landslide entered accelerative deformation stage. In the meanwhile, a thorough collection of the macroscopic proofs also suggests that historical maxima are highly consistent with landslide macroscopic deformation behaviours.

Compared with traditional safety factor, state fusion entropy evaluates the landslide instability, and is capable to indicate its extent and changing regularities. Compared with simulation methods for landslide stability analysis, this approach takes

displacement monitoring data as the basis of landslide stability analysis, and thus is prone to continuous stability analysis. Compared with direct judgment from displacement monitoring data, this approach analyse landslide deformation states by a data-driven model, avoiding the disunity of individual engineering geology experience, ensuring its applicability to the geological conditions of different landslides.

5    However, several issues also need to be clarified. Firstly, data selection and feature extraction are simplified. Although monitoring data of multi-point and multi-sensor are helpful to express the comprehensive state of landslide, relevant research is still in progress and a common practice, selecting one typical displacement data of GPS, is adopted for now. Besides, displacement data is currently obtained monthly by GPS. At this time scale, deformation velocity and acceleration are considered to express landslide deformation well and thus selected for deformation state definition. For higher time resolution

10   data, some feature extraction methods may be necessary to determine the DE and DT indexes. Finally, Entering into accelerative deformation stage is a necessary condition for landslide failure. Aiming at this, the fluctuant increasing tendency of cumulative state fusion entropy and the frequent renewal of historical maximum may help to judge whether landslide has entered accelerative deformation stage or not. Once this happens, other clues such as macro cracks should also be taken into account to fully determine landslide early warning level. In addition, the Markov property of deformation state can be used for

15   prediction.

Technical corrections

1.    pag. 1 line 13: the evolutionary stages of the phenomenon Modified

2.    pag. 1 line 15: for assessing landslide stability Modified

20   3.    pag. 1 line 18: damages of properties every year Modified

4.    pag. 1 line: at site specific scale Modified

5.    pag. 2 line 5: it becomes of interest to find Modified

6.    pag. 2 line 9: Due to its easy acquisition Modified

7.    pag. 2 line 16: Previous works have introduced Modified

25   8.    pag. 3 line 8: the individuation of different deformations states Modified

9.    pag. 3 line 9: to investigate deformation states Modified

10.   pag. 7 line 18: As time goes on Modified

11.   pag. 8 line 4: with a length of 2000 m Modified

12.   pag. 8 line 17: monthly indexes for Xintan landslide Modified

30   13.   pag. 13 line 8: entering accelerative deformation stage highlighted in previous works (please insert references about this) Modified

**Response to Anonymous Referee #2:**

1. Nevertheless, I find that this paper does not really show whether the proposed approach gives a real advantage over other existing data-driven, empirical or physically based methods in quantifying landslide stability/instability. A comparison of several methods would be greatly helpful.

In introduction, the basic principles and main advantages and disadvantages of the existing methods (Saito's method, FEM, LEM) are detailed, hoping to highlight the starting point of this paper. In the case study, it is difficult to compare the results of this method with other methods whose results usually are presented with safety factor, because this paper indicates landslide instability with the proposed state fusion entropy index. As a supplement, more macroscopic phenomenon has been added as the evidence to validate the effectiveness of this method.

Saito's method is an empirical forecast model and is suitable for the prediction of sliding tendency and then the failure time. Based on homogeneous soil creep theory and displacement curve, it divides displacement creep curves into three stages: deceleration creep, stable creep and accelerating creep, and establishes a differential equation for accelerating creep. The physical basis of Saito's method helped it to successfully forecast a landslide that occurred in Japan in December 1960, but also makes it strongly dependent on field observations. LEM is a kind of calculation method to evaluate landslide stability based on mechanical balance principle. By assuming a potential sliding surface and slicing the sliding body on the potential sliding surface firstly, LEM calculates the shear resistance and the shear force of each slice along the potential sliding surface and defines their ratio as the safety factor to describe landslide stability. LEM is simple and can directly analyse landslide stability under limit condition without geotechnical constitutive analysis. However, this neglect of geotechnical constitutive characteristic also restricts it to a static mechanics evaluation model that is incapable to evaluate the changing regularities of landslide stability. In the meanwhile, LEM involves too many physical parameters such as cohesive strength and friction angle, which makes it greatly limited in landslide forecast and early warning. As a typical numerical simulation method, FEM subdivides a large problem into smaller, simpler parts that are called finite elements. The simple equations that model these finite elements are then assembled into a larger system of equations that models the entire problem. FEM then uses variational methods from the calculus of variations to approximate a solution by minimizing an associated error function. In landslide stability analysis, FEM can not only satisfy the static equilibrium condition and the geotechnical constitutive characteristic, but also adapt to the discontinuity and heterogeneity of the rock mass. However, FEM is quite sensitive to various involved parameters and the computation will increase greatly to get more accurate results. If parameters and boundaries are precisely determined, LEM and FEM can provide results with high reliability. [Has been added in "Introduction"]

The macroscopic behaviours of Xintan landslide near historical maxima were investigated according to previous studies (Wang, 1996). In June 1982, some trees in the top area of Jiangjiapo were dumped. A small amount of north-west tensile cracks appeared on the steeper section of the east. Around August 1982, the front edge of Jiangjiapo went through a small collapse. In June 1983, the colluvial deposits between Guangjiaya and Jiangjiapo showed signs of resurrection. At the end of 1984, the trailing edge of the landslide showed an "armchair" shape and the leading edge was bulged out. Some collapse pits were found

on the upper side while several new tensile cracks in the middle. Meanwhile, some small collapses which seem irrelevant to rainfall occurred. In May 1985, old cracks widened and new cracks appeared, forming a ladder-shaped landing ridge. Moreover, Jiangjiapo presented a clear trend of the overall slippage. These proofs suggest that the historical maximum index is highly consistent with landslide macroscopic deformation behaviours. [Has been added in "Case study"]

5   2.   Furthermore, there is no evidence that the method can be successfully used in an early-warning perspective, which is the goal set in the abstract. My main concern is that the entropy approach used by the authors is based solely on measurements of displacements, seemingly in a single point of a landslide. The authors show that the pattern of state fusion entropy is (not surprisingly!) consistent with that of displacements (input information). Thus, what does the entropy tell in addition to what is already obvious by looking at the displacement pattern and, perhaps, by setting displacement rate thresholds

10      to provide early warning? This has not been clarified. In addition, can the performance of the model be improved by integrating several displacement measurements (and perhaps pore pressures, water level, water content, deep deformations, etc.)? This is an important topic to be addressed.

Thanks for your comments. First of all, state fusion entropy and displacement may be similar but can also be different. This has been pointed out in the explanation of Figure 8 but may seems not obvious enough. Corresponding paragraph has been

15 rephrased. There is no doubt that monitoring data of multi-point and multi-sensor contain the information about landslide state and much more comprehensive landslide states can be obtained if all these monitoring data are utilized. However, this comprehensive monitoring data is not yet common. And thus a traditional operation, selecting one typical displacement data of GPS, is adopted for generality and simplicity.

Similarities and differences between displacement and state fusion entropy are found through a comparative analysis of these

20 landslides. As for Bazimen landslide and Pajiayan landslide, cumulative state fusion entropy and cumulative displacement show similar change rules especially during the drawdown period of water level, indicating their intrinsic consistency. As for Baishuihe landslide and Shuping landslide, cumulative state fusion entropy of shows a distinctly different characteristic from their cumulative displacement. Taking Baishuihe landslide as an example, the severe deformation in June 2007 seems to suggest that the landslide has entered accelerative deformation stage. However, subsequent monitoring has proved that the

25 deformation is only a temporary effect of heavy rainfall and fluctuation of water level (Xu et al., 2008). In Figure 8, cumulative state fusion entropy of Baishuihe landslide returns to a low level after several historical maxima. [Has been rephrased in "Case study"]

Although monitoring data of multi-point and multi-sensor are helpful to express the comprehensive state of landslide, relevant research is still in progress and a common practice, selecting one typical displacement data of GPS, is adopted for now. [Has

30 been added in "Conclusion and Discussion"]

3.   It may be argued that the displacement rate thresholds are set arbitrarily in a displacement-based monitoring system. However, I see that even in this data-driven approach there are arbitrary site-specific decisions made by the authors (e.g. page 9 line 4), which perhaps can affect the model output. So, for a model to be truly data driven, I expect no arbitrary choices, or arbitrary choices to have little influence: the dataset should provide the answer itself.

Thanks for your advice. Theoretically, the k-means clustering method is based on the data distribution of input data. The cluster number K only determines the division roughness of clusters and has little impact on the distribution of clusters which is the basis of the state fusion entropy approach. Therefore, the cluster number was empirically set to 3 in the case study. Now some strategies have been proposed to determine cluster number totally and automatically according to input data. And this can also

5   be used as an improvement of the method.

4.   Finally, the content of the work does not seem to match its title: monthly displacements are probably too far from a "real-time" landslide monitoring when incipient failure is concerned. I expected to see interpretation of daily, hourly or even more frequent observations of landslide displacements prior to failure.

Thanks for your constructive suggestion. "real-time" has been replaced with "continuous" in title and text. In the meanwhile,

10   the description of the inputs of the joint clustering method has been modified to avoid confusion with the ($v$, $a$) in case study. Moreover, the reason why deformation velocity and acceleration are selected in case study has been addressed. This method is capable for higher time resolution data, but the data features may need to be determined by some feature extraction methods. To describe clearly this method, two functional data type are defined for landslide displacement data. One is to indicate deformation extent (*DE*) and the other to deformation tendency (*DT*). Positive *DT* indicates an increasing deformation. The

15   process of defining deformation states is as follows. Step 0. Unite DE and DT at the same time as an item, i.e., (*DE*, *DT*); [Has been added in 2.1]

Considering that the monitoring error of GPS can be ignored compared to landslide actual deformation on monthly time scale, monthly deformation velocity ($v$) was selected as the *DE* index and monthly deformation acceleration ($a$) as the *DT* index. [Has been added in Case study]

20   Besides, displacement data is currently obtained monthly by GPS. At this time scale, deformation velocity and acceleration are considered to express landslide deformation well and thus selected for deformation state definition. For higher time resolution data, some feature extraction methods may be necessary to determine the *DE* and *DT* indexes. [Has been added in "Conclusion and Discussion"]

5.   Due to these concerns, I feel that this manuscript is not ready for publication in the present form. I recommend the authors

25   update their work by addressing the above points and, in particular, by including evidence of good performance of their model in making usable predictions of landslide failure based on high-resolution displacement patterns, which could be used in an early warning system.

Entering into accelerative deformation stage is a necessary condition for landslide failure. Aiming at this, the fluctuant increasing tendency of cumulative state fusion entropy and the frequent renewal of historical maximum may help to judge

30   whether landslide has entered accelerative deformation stage or not. Once this happens, other clues such as macro cracks should also be taken into account to fully determine landslide early warning level. In addition, the Markov property of deformation state can be used for prediction. [Has been added in "Discussion and conclusion"]

**Response to Anonymous Referee #3:**

1.  Firstly, the advantages and the limitations of existing methods seems too brief to emphasize the meaning and emergency of the proposed approach. The processes of the model is complex, please organize this part clearly. I suggest that the methods should be divided into several subsections. This method named "the proposed joint clustering method combining k-means and cloud model" should be refined. The part of "materials and results" should be correspondence with the part of "methods".

Thanks for your kind suggestion. Firstly, a detailed introduction to these methods (Saito's method, LEM and FEM) has been added, including their advantages and the limitations. Given that several methods are involved in this approach, we have tried our best to divide it into two near-independent parts, respectively the definition and the multi-attribute entropy analysis of deformation states. Too much sub-section may undermine the integrity of the content. The method name "the proposed joint clustering method combining k-means and cloud model" may be too long but it expresses apparently the essential factors of this method. K-means and cloud model complement each other, together form the core of the joint clustering. Sorry, we have not figure out a better alternative. Any suggestions and advices on this issue are always welcome.

Saito's method is an empirical forecast model and is suitable for the prediction of sliding tendency and then the failure time. Based on homogeneous soil creep theory and displacement curve, it divides displacement creep curves into three stages: deceleration creep, stable creep and accelerating creep, and establishes a differential equation for accelerating creep. The physical basis of Saito's method helped it to successfully forecast a landslide that occurred in Japan in December 1960, but also makes it strongly dependent on field observations. LEM is a kind of calculation method to evaluate landslide stability based on mechanical balance principle. By assuming a potential sliding surface and slicing the sliding body on the potential sliding surface firstly, LEM calculates the shear resistance and the shear force of each slice along the potential sliding surface and defines their ratio as the safety factor to describe landslide stability. LEM is simple and can directly analyse landslide stability under limit condition without geotechnical constitutive analysis. However, this neglect of geotechnical constitutive characteristic also restricts it to a static mechanics evaluation model that is incapable to evaluate the changing regularities of landslide stability. In the meanwhile, LEM involves too many physical parameters such as cohesive strength and friction angle, which makes it greatly limited in landslide forecast and early warning. As a typical numerical simulation method, FEM subdivides a large problem into smaller, simpler parts that are called finite elements. The simple equations that model these finite elements are then assembled into a larger system of equations that models the entire problem. FEM then uses variational methods from the calculus of variations to approximate a solution by minimizing an associated error function. In landslide stability analysis, FEM can not only satisfy the static equilibrium condition and the geotechnical constitutive characteristic, but also adapt to the discontinuity and heterogeneity of the rock mass. However, FEM is quite sensitive to various involved parameters and the computation will increase greatly to get more accurate results. If parameters and boundaries are precisely determined, LEM and FEM can provide results with high reliability. [Has been added in "Introduction"]

2. Secondly, in the "Deformation state definition based on K-means combined with Cloud Model", a better explanation why deformation rate and acceleration are selected to define deformation states may be necessary. How the displacement data was chosen because it is quite common for a landslide to have multiple displacement monitoring points at present.

Thanks for your kind advice. The description of the inputs of the joint clustering method has been modified to avoid confusion with the ($v$, $a$) in case study. Moreover, the reason why deformation velocity and acceleration are selected in case study has been addressed.

To describe clearly this method, two functional data type are defined for landslide displacement data. One is to indicate deformation extent (*DE*) and the other to deformation tendency (*DT*). Positive *DT* indicates an increasing deformation. The process of defining deformation states is as follows. Step 0. Unite *DE* and *DT* at the same time as an item, i.e., (*DE*, *DT*); …
[Has been added in 2.1]

Considering that the monitoring error of GPS can be ignored compared to landslide actual deformation on monthly time scale, monthly deformation velocity ($v$) was selected as the *DE* index and monthly deformation acceleration ($a$) as the *DT* index. [Has been added in Case study]

Although monitoring data of multi-point and multi-sensor are helpful to express the comprehensive state of landslide, relevant research is still in progress and thus a common practice, selecting one typical displacement data of GPS, is adopted for now. [Has been added in "Conclusion and Discussion"]

3. Thirdly, in the "materials and results" section, only monthly displacement data was used and it seems not very consistent with "real-time" in the title. Since for now monthly monitoring displacement is mainly adopted in most studies, "monthly stability" may be more appropriate for the title. In the meanwhile, the discussion on the process of other monitoring frequency data needs to be added.

Thanks for your kind suggestion. "real-time" has been replaced with "continuous" in title and text. Discussion on higher time resolution data has been added.

Besides, displacement data is currently obtained monthly by GPS. At this time scale, deformation velocity and acceleration are considered to express landslide deformation well and thus selected for deformation state definition. For higher time resolution data, some feature extraction methods may be necessary to determine the *DE* and *DT* indexes. [Has been added in "Conclusion and Discussion"]

4. Finally, "Discussion" and "Conclusion" present several repetitions and need a better description. Meanwhile, the English written of this paper should be modified carefully again.

Thanks for your constructive suggestion. We have merged and rephrased the "Discussion" and "Conclusion".

**The list of all relevant changes made in the manuscript**

Notice: all the page number and line number refer to that in the marked-up manuscript.

Page 1 line 1, Page 1 line 7, page 2 line 23, page 3 line 8: replaced "real-time" with "continuous" in the title

Page 1 line 15: replaced "judging" with "assessing"

5   Page 1 line 17: replaced "causalities and property damage" with "damages of properties"

Page 1 line 19: replaced "for" with "as"

Page 1 line 21- page 2 line 9: detailed the introduction to methods (Saito's method, LEM and FEM), including their advantages and the limitations.

Page 2 line 13, page 5 line 8: replaced "can't" with "cannot"

10  Page 2 line 20: replaced "Then comes the interest" with "It becomes of great interest"

Page 2 line 24: replaced "Benefits from" with "Due to"

Page 2 line 26-32: added the references about displacement thresholds and their values

Page 3 line 2: replaced "Some scholars" with "Previous works"

Page 3 line 5: added the references about the application of entropy to landslide stability analysis

15  Page 3 line 12: replaced "Methodologies" with "Methods"

Page 4 line 7: replaced "excavate" with "investigate"

Page 5 line 11: rephrased the description of the inputs of the joint clustering method

Page 6 line 16, page 7 line 29: replaced "individual contribution" with "individuation"

Page 8 line 16: replaced "Materials and Results" with "Case study"

20  Page 8 line 18-20: detailed the landslide type of studied landslides

Page 8 line 23: replaced "about 2000 m long" with "with a length of 2000 m"

Page 9 line 3: replaced "of" with "for"

Page 9 line 8-9: added the explanation why monthly deformation velocity and acceleration were selected

Page 12 line 13- page 13 line 2: added the macroscopic phenomenon to suggest the effectiveness

25  Page 13 line 17: rephrased this paragraph to indicate the similarities and differences between SFE and displacement

Page 13 line 21: rephrased this sentence

Page 14 line 3: merged "Discussion" and "Conclusion"

Page 14 line 4-13: rephrased the main work of this study

Page 14 line 14-page 15 line 2: rephrased the advantages of this method

30  Page 15 line 3-13: rephrased the discussions on data & feature selection and potential application in early warning system

**The marked-up manuscript**

[revised manuscript text omitted]

批注 [Z4]: Response to #1
General comments 1
Response to #2
General comments 1
Response to #3
General comments 1

批注 [Z5]: Response to #1
Specific comments 2

批注 [Z6]: Response to #1
Technical comments 5

批注 [Z7]: Response to #1
Technical comments 6

批注 [Z8]: Response to #1
General comments 2
Specific comments 1

[revised manuscript text omitted]

批注 [Z12]: Response to #1
Specific comments 2

批注 [Z13]: Response to #1
Specific comments 2

批注 [Z14]: Added
To avoid confusion with (v, a) in case study

[revised manuscript text omitted]

批注 [Z17]: Response to #1
Technical corrections 10

批注 [Z18]: Response to #1
Specific comments 2

批注 [Z19]: Response to #1
Specific comments 5

批注 [Z20]: Response to # 1
General comments 3

批注 [Z21]: Response to #1
Technical corrections 11

[revised manuscript text omitted]

批注 [Z27]: Response to #1
    Specific corrections 7
Response to #3
    General comments 4

批注 [Z28]: Response to #1
    General corrections 3
Response to #2
    General corrections 5

[revised manuscript text omitted]

---

## Author Response (AR2)

**The point to point response to the reviewers**

First of all, we would like to express our sincere appreciation of your very constructive comments and suggestion.

Next, in a sequence, we would like to respond to comments in a point to point manner so that hopefully all the questions can be answered or clarified. All the responses are in red and all the changes made in the manuscript are underlined.

**Response to Anonymous Referee #1:**

The Authors improved significantly the overall quality of the paper, answering to all the revisions required by the Anonymous Referees. Methods and results are explained in a better way. Discussions about the results of the research are also clearer and help in the comprehension of the goals and of the achieved results of the research. Instead, a comparison, in terms of advantages and limits, between the proposed methodology and the other described methods (Saito's method, FEM, LEM) can be added in the Discussions and Conclusions section, for improving the analysis about this new approach.

Res:we have tried to find applications of methods like LEM and FEM in the research of Xintan landslide. But only several Chinese literatures are found and has been added. In addition, the result of an unloading-loading response ratio method (ULRR) is added for non-Chinese readers. We hope this can be helpful for results comparison.

In addition, the results of several other studies were introduced for comparison. Chen (2014) studied the stability of Xintan landslide by FEM with consideration of the loading effect and material weakening caused by rainfall, as shown in Figure 8. Moreover, the result of an unloading-loading response ratio method (ULRR) is also introduced (Zhang et al., 2006; He et al., 2010), as shown in Figure 8. Because only annual results are given in these studies, annual average of CSFE were correspondingly calculated for comparison. According to Figure 8, the safety factor decreases year by year and cannot reflect the recovery process of landslides stability. The ULRR presents similar changing regularities like CSFE after Xintan landslide entered accelerative deformation stage in 1982. But the mutation in 1981 when Xintan is still in uniform deformation stage seems unreasonable. Besides, ULRR is obtained yearly and offers less details about stability changes than CSFE. [Has been added in Case study]

[Figure]

Figure 8. Comparison of CSFE with ULRR and safety factor of Xintan landslide

**Response to Anonymous Referee #2:**

The reviewer appreciates the effort put by the authors in replying to the remarks and working on some of them. However, many of the doubts and issues raised by the reviewer remain unsolved. Therefore, the reviewer suggests that the manuscript should not be accepted for publication in the present form, but encourages resubmission after all the following points being solved or, at least, discussed appropriately.

1) A comparison of several methods to provide evidence that the proposed model does give a practical advantage over the existing methods for the "continuous and site-specific analysis of landslide stability changing regularities". The reviewer expects to see a comparison with the methods mentioned by the authors (Saito's, FEM, LEM) in at least one case study.

Res:We have tried to find applications of methods like LEM and FEM in the research of Xintan landslide. But only several Chinese literatures are found and has been added. In addition, the result of an unloading-loading response ratio method (ULRR)

is added for non-Chinese readers. We hope this can be helpful for results comparison.

In addition, the results of several other studies were introduced for comparison. Chen (2014) studied the stability of Xintan landslide by FEM with consideration of the loading effect and material weakening caused by rainfall, as shown in Figure 8. Moreover, the result of an unloading-loading response ratio method (ULRR) is also introduced (Zhang et al., 2006; He et al., 2010), as shown in Figure 8. Because only annual results are given in these studies, annual average of CSFE were correspondingly calculated for comparison. According to Figure 8, the safety factor decreases year by year and cannot reflect the recovery process of landslides stability. The ULRR presents similar changing regularities like CSFE after Xintan landslide entered accelerative deformation stage in 1982. But the mutation in 1981 when Xintan is still in uniform deformation stage seems unreasonable. Besides, ULRR is obtained yearly and offers less details about stability changes than CSFE. [Has been added in Case study]

[Figure]

Figure 8. Comparison of CSFE with ULRR and safety factor of Xintan landslide

2) Application to early warning: the authors state in the abstract that "state fusion entropy may serve as a novel index for assessing landslide stability and landslide early warning". The reviewer expects to see, in the revised manuscript, an example of application of the proposed method to the early warning of a landslide (a past case study is fine), showing a successful (and usable) prediction of failure.

Res: The main purpose of this paper is to obtain landslide stability changing regularities by performing multi-attribute entropy analysis on displacement monitoring data. Given the close relationship between stability and early warning, SFE also has the potential to be applied in early warning. But for now, SFE has not been intended to be applied to landslide early warning unless be qualified by further research. In order to avoid misunderstandings, the "early warning" mentioned in the article will be modified.

3) Integration of multiple monitoring sources. The reviewer expects to see an example of performance of the model by taking displacement data from different locations as input (or at least a good discussion about this). Multiple monitoring, to the reader's opinion, is already a diffused standard in landslide monitoring, and must be taken into account when proposing a "novel method". The reviewer also expects a deeper discussion on the incorporation of different monitorable quantities (e.g. pore pressures, moisture, deep displacements, etc.).

Res: Multiple monitoring indeed has already become a common practice in landslide monitoring, but the comprehensive mining of multi-source data is also still a common problem. Results and conclusions about this issue may be difficult for now, but we do already have some thoughts. For the open system of landslide, the displacement of different monitoring points can be regarded as landslide samples with different deformation scales. Meanwhile, fractal theory tells that same patterns as the entire system can be found if a small part of the whole be magnified. Therefore, fractal theory is intended to be introduced for multi-point data analysis.

4) Arbitrary choices. The reviewer expects to see in the revised manuscript an analysis of the effect of arbitrary choices on the model's performance. For instance, how does the result changes if a different number of clusters is used?

Res: Theoretically, the k-means clustering method is based on the data distribution of input data. The cluster number K only determines the division roughness of clusters and has little impact on the distribution of clusters which is the basis of the state fusion entropy approach. To prove this point, CSFE with different cluster numbers (K=3 to 7) has been added and discussed.

To measure the influence of different cluster numbers on the performance of this method, CSFE with different cluster numbers (K=3 to 7) is compared, as shown in Figure 10. As can be seen, CSFE varies slightly with K. This is mainly because that different K correspond to different division roughness of deformation states, which sequentially affects the value of the CSFE. Due to this fluctuation, CSFE is not intended to be applied to landslide early warning unless be qualified by further research. Despite this, the overall trend remains unchanged, which suggests a steady statistical regularity of deformation states. [Has been added in Discussion and conclusion]

[Figure]

**Figure 10. CSFE with different cluster numbers (K=3 to 7) of Xintan landslide**

5) Real-time analysis. The reviewer appreciates that the authors changed the title from "real-time" to "continuous". However, by using monthly data, early warning in acceleration phase seems unlikely. Hence, the authors are suggested to eliminate all references to early warning. Alternatively, they should use daily or, better, hourly / minutely input to run the model. During the acceleration phase of a movement, the input frequency should be increased if a real-time (or a continuous) monitoring for early warning purposes is desired. Moreover, landslides moving at mm/day or mm/week rates are certainly monitorable with daily frequency without significant error, if proper monitoring is used (e.g. high-quality GPS measurements, inclinometer measurements; examples in the literature are available), so the reviewer does not understand why the authors insist on using monthly measurements. The authors could easily apply the model to a different case study, where better monitoring data series are available, to prove the good performance of their model.

Res: We are also looking forward to displacement monitoring data with higher time resolution for more in-depth research. But unfortunately, we do not have that yet. Therefore, we will take your suggestion to adjust the word "early warning" in the paper.

6) Smoothed CSFE. I have severe doubts on the use of a smoothed CSFE in the work, as it seems to me that the authors erroneously used a 5-period average obtained from two past data, the current value and two future data. The future data are obviously unavailable during real-time monitoring, so the SCSFE they obtained simply makes no sense! I suggest the authors avoid any smoothing, as the period is chosen arbitrarily. Moreover, if monthly data are already very far from being a "real-time" monitoring, a 5-months average value (of the past data) is even farther. Without the smoothing, it becomes also clear from the figure the authors present that the CSFE is unable to predict the failure: in the case of the Xintan landslide (fig. 7), failure occurred at a value of CSFE which is lower than the historical maximum. If a warning threshold was set to correspond to the CSFE overcoming its past maximum, you would have failed to warn the authorities/population before failure occurred. Talking about Fig. 7, you did not include the displacement trend of the landslide, despite my explicit request. I believe this is fundamental (as done for fig. 8), to let the reader compare your statistics with the actual monitoring data.

Res: Thanks for your constructive suggestion. We have realized the problem of the 5-window smoothing. Historical maxima will be judged directly the CSFE without any smoothing. At the same time, a clarification for the "failure in prediction" is quite necessary. CSFE does not predict the failure of landslide but provides information about stability changing regularities. Besides, the huge displacement in June 1985 when Xintan failed has not been adopted in this SFE analysis because it's believed that field surveys are much more important than numerical judgments when the landslide has shown obvious signs of slipping.

**Response to Anonymous Referee #3:**

The quality of the paper was improved during the review process, addressing (or answering) to all the comments made by the Reviewers. The manuscript still needs a discussion on the quantitative/qualitative comparison between the proposed methods and other methods - as Saito's method, FEM, or LEM - described in the paper. This should be added in the discussion before the paper can be considered acceptable for publication.

Res:We have tried to find applications of methods like LEM and FEM in the research of Xintan landslide. But only several Chinese literatures are found and has been added. In addition, the result of an unloading-loading response ratio method (ULRR) is added for non-Chinese readers. We hope this can be helpful for results comparison.

In addition, the results of several other studies were introduced for comparison. Chen (2014) studied the stability of Xintan landslide by FEM with consideration of the loading effect and material weakening caused by rainfall, as shown in Figure 8. Moreover, the result of an unloading-loading response ratio method (ULRR) is also introduced (Zhang et al., 2006; He et al., 2010), as shown in Figure 8. Because only annual results are given in these studies, annual average of CSFE were correspondingly calculated for comparison. According to Figure 8, the safety factor decreases year by year and cannot reflect the recovery process of landslides stability. The ULRR presents similar changing regularities like CSFE after Xintan landslide entered accelerative deformation stage in 1982. But the mutation in 1981 when Xintan is still in uniform deformation stage seems unreasonable. Besides, ULRR is obtained yearly and offers less details about stability changes than CSFE. [Has been added in Case study]

[Figure]

Figure 8. Comparison of CSFE with ULRR and safety factor of Xintan landslide

**The list of all relevant changes made in the manuscript**

Page 1 line 15: changed "landslide early warning" to "judging landslide evolutionary stages"

Page 1 line 19: removed "and early warning"

Page 2 line 22: removed "and early warning"

Page 3 Figure 1: Changed "landslide early warning" to "evolutionary stage judgment"

Page 3 line 11: changed "for landslide early warning" to "to identify key time nodes of stability changes"

Page 8 line 9: changed "For landslide early warning" to "Besides"

Page 8 line 11: removed "smoothed"

Page 8 line 11-12: removed "which is conducted mainly based on the consideration that stability cannot be mutant before landslide failure"

Page 12 Figure 7: removed SCSFE, represented historical maxima with red column

Page 12 line 4: removed "and average-smoothed with a window of 5"

Page 13 line 8: removed ", which also corresponds to a new historical maximum"

Page 13 line 13-20: added the results comparison of CSFE

Page 14 Figure 8: added the figure of results comparison

Page 15 line 8: changed "landslide early warning" to "identifying key time nodes of stability changes"

Page 15 Figure 9: removed SCSFE, represented historical maxima with red column

Page 16 line 3-7: added the discussion of the influence of different cluster numbers on the CSFE

Page 16 Figure 10: added toe figure of CSFE with different cluster numbers

Page 17 line 1: removed "In addition, the Markov property of deformation state can be used for prediction."

Page 17 line 21-22: added relevant reference in results comparison

Page 17 line 38-40: added relevant reference in results comparison

Page 19 line 15-16: added relevant reference in results comparison

**The marked-up manuscript**

[revised manuscript text omitted]